# Multi-Criteria Analysis of the "Lake Baikal—Irkutsk Reservoir" Operating Modes in a Changing Climate: Reliability, Resilience, Vulnerability



**Alexander Buber** [1,*] **and Mikhail Bolgov** [2]

1 All-Russian Research Institute of Hydraulic Engineering and Land Reclamation named after A.N. Kostyakov, 44, B. Akademicheskaya Str., 127550 Moscow, Russia
2 Water Problems Institute, Russian Academy of Sciences, 3, Gubkina Str., 119333 Moscow, Russia; bolgovmv@mail.ru
* Correspondence: buber49@yandex.ru

**Abstract:** In the second half of the twentieth century, a cascade of reservoirs was constructed along the Angara: Irkutskoe, Bratskoe, Ust-Ilimskoe and Boguchanskoe, which were intended for producing renewable hydroelectric energy for providing transportation through the Angara and Yenisei Rivers, and for avoiding floods. The upper reservoir (Irkutsk Dam) is used to regulate the level of Baikal Lake. The cascade of Angarsk reservoirs is managed using the dispatch schedules developed in 1988. This article contains a multi-criteria analysis of the "Lake Baikal–Irkutsk Reservoir" operating modes in a changing climate, based on statistical summaries of performance criteria: reliability, resilience, vulnerability. Studies have shown that dispatch schedules need to be developed on the historical series of recent years, updated more often and optimization methods should be used for real-time management. This article discusses mathematical methods, algorithms and their implementations for the formation of reservoir operation modes based on dispatch schedules (DS) and optimization methods. Furthermore, mathematical methods, algorithms and programs have been developed for the formation of reservoir operation modes in real time, based on optimization approaches and long-term series of observed inflows, taking into account a given hierarchy of priorities of water users' requirements. To solve the integer nonlinear large-dimensional task of performing water resource calculations, a special optimization algorithm was developed that allows decomposition of the task into a series of two-year dimensional independent subtasks.

**Keywords:** water resource calculation; dispatch schedule; time series of the inflow; release rules; reliability; resilience; vulnerability; optimization methods; multi-criteria analysis; trade-offs solution





## 1. Introduction

Baikal is the largest freshwater lake in the world, it contains more than 20% of the world's fresh surface water. The maximum depth of Lake Baikal is more than 1600 m (5387 feet). Baikal is considered to be one of the cleanest lakes in the world. Lake Baikal was declared a UNESCO world heritage site in 1996.

The operational management of reservoir operation modes in Russia is carried out using dispatch schedules (DS), developed on the basis of long-term hydrological series of the observed inflow and statistical analysis methods. Management of the "Lake Baikal–Irkutsk Reservoir" complex is based on the use of dispatch schedules developed in 1988. Due to the changing environment, most likely related to climate warming, the dispatch schedules are hopelessly outdated and although there have been attempts to develop new rules in 2007 and 2013, they have not been approved and require constant improvement.

The main disadvantage of this approach is the inability to consider climatic fluctuations without changing the dispatch schedules themselves, reflecting some average reliability indicators without considering changes in the hydrological situation in recent

years. Studies of operating methods, based on the dispatch schedule's parameters optimization, are described in [1–3].

The purpose of the research presented here is a multi-criteria analysis of the Irkutsk reservoir operating modes under different hydrological conditions, based on dispatch schedules and optimization methods. The paper presents mathematical methods, algorithms and computational technologies for the formation of reservoir operation modes, allowing one to consider the long-term interval hydrological inflow series observations and the water users' requirements priorities hierarchy. The research was carried out on the basis of the stochastic hydrology method use, optimization methods and the theory of making compromised decisions, set out in [4] (Chapter 9).

The performed studies are based on three fundamental statistical criteria used to assess the possible performance of a water resource system (WRS): reliability, resilience, vulnerability. The first is a system failure probability, the second estimates how quickly the system recovers after failure and the third shows the seriousness of consequences in case of failure. For the first time, these criteria were strictly defined in work [5], in which it was recommended to use them to perform a multi-criteria analysis of the WRS, to justify the developed release rules and strategic planning in the climate change. Works [6,7], which mention many similar studies by other authors, are characteristic of this type of research.

Many authors in their studies, especially in recent years, have used various modifications of these criteria, replacing the average parameters with the maximum ones. So, article [6] provides a review of the reliability, resilience and vulnerability (RRV) estimators proposed in the literature, and examines which combination of these would be the most appropriate for use in connection with a multi-objective risk assessment and the sustainability of a scenario for the development of a WRS. Different definitions of resilience and vulnerability values based on either mean or maximum deviations from satisfactory states, were discussed. Estimation based on historical time series is shown to be problematic, and a procedure encompassing generation of synthetic time series with a length of at least 1000 years is recommended in order to stabilize the estimates. Moreover, the strong correlation between resilience and vulnerability may suggest that resilience should not be explicitly accounted for.

Although the authors justified performing water resource calculations for the "Lake Baikal–Irkutsk Reservoir" complex using the dispatch schedule (DP) on artificial 10,000-year series, in the current studies, however, where the necessity of using optimization methods is proved, it was necessary to limit them to a 44-year series, for which optimization can be conducted. In addition, as practice has shown, since hydrological series of large dimensions are formed on the basis of historical series statistical parameters, the results of calculating reliability criteria differ little from calculations based on historical series. Therefore, the hypothesis in [6] about the need for calculations for 1000 synthetic series is extremely problematic. The natural correlation between resilience and vulnerability is also insignificant. These criteria provide a qualitative and quantitative assessment of failures, so the authors actively used both criteria.

Various modifications do not give significant advantages, therefore, in this article, we used the classical definitions of the criteria given in [4] (Chapter 9). In contrast to [5,6], the studies paid special attention to considering WRS with a large number of often conflicting water users' requirements. For this, the normalized vulnerability and the integrated were determined by a set of criteria, Normalized Reliability Index and Integrated Normalized Vulnerability Index.

Article [7] describes a developed systematic procedure Tai WAP to quantify changes in water resources and improvements after implementing adaption measures. Assessing water resources vulnerability and resilience are performed using an integrated tool, including climate change scenarios, a weather generator, a hydrological model and system dynamic models (unfortunately, the release rules used in the procedure are not given). Since the generators of the climate change scenarios formation indicated in [7] poorly predict the future hydrological situation, especially for regions where the inflow depends

on precipitation, snow cover and glaciers melting, the study decided to make a forecast for the historical inflows' series from 1903 to 2020, dividing it into two parts: average water content (1903–1994) and low water content (1995–2020). For these two scenarios, a 44-year hydrological forecast was carried out.

It should be noted that in many of the studies mentioned in [6,7], simplified rules for controlling releases are used, which significantly reduce the confidence in the results and conclusions. Therefore, the authors used real release rules (dispatch schedules) with a complex computational scheme for the reservoir operation modes formation. In the conducted studies (in contrast to [6,7]), on the basis of statistical criteria, the nature of hydrological changes that influenced the fulfillment of water users' requirements was determined, and the reliability of the WRS was assessed when managed in accordance with the current release rules.

In almost all studies [8–12], optimization methods were used for a comparative assessment of the existing management rules' reliability, or for the promising measures formation to improve the water management situation in the face of climate change (strategic planning).

So, in study [8], an inexact two-stage stochastic programming model was developed for The Yinma River Basin supporting water resources allocation. The planning horizon covers 15 years and has three scenarios for flow levels (low, medium and high). The model makes it possible to form the optimal distribution of water resources between the four main water use sectors (industry, municipality, ecology and agriculture) based on the adopted scenario of climate change, and serves as a strategic planning tool

In the study [9], an optimal allocation model for a large complex system of water resources was developed by considering both water supply and river ecological benefits. The water supply benefit was defined as the minimum water deficit for different water users, while the ecological benefit involves making the reservoir release as close as possible to the natural streamflow. The objective functions were normalized and transformed into non-dimensioned variables ranging from 0 to 1. To solve this problem, the combination of decomposition-coordination (DC) and discrete differential dynamic programming (DDDP) methods were proposed. The decomposition in [9], performed by region, significantly reduced the optimization time, however, it can lead to an imbalance in the direction of one of the subsystems and the general solution may be far from optimal. For example, optimization for a region with a reclamation or industry and a region with water supply generates a conflict of interest and requires a multi-criteria analysis. Therefore, in our study, the decomposition was performed two years over the entire long-term series, with a cyclic repetition during the optimization process. This allows the use of multiprocessor and multitasking computer mode in the solving process, which reduces the calculation time by tens of times. In addition, two-year series optimization can be performed for the entire WRS using powerful optimization platforms (such as Solver).

In studies [8,9], unfortunately, the adopted (traditional) rules for reservoir management are poorly formulated. However, when planning measures to improve the state of the WRS, the main result can be achieved through the development of effective management rules that can compensate for the water deficit arising in the context of climate change. A multi-criteria analysis of the water resources availability has not been done according to statistical criteria of efficiency (reliability, resilience, vulnerability). The analysis was carried out using the values of the objective function (OF), which do not reflect the water availability of the WRS. The stated research methodology is intended for strategic planning in the context of climate change, but does not allow the decision maker (DM) to make operational decisions on the reservoir water resources management. In contrast to [8,9], the main emphasis was placed on the practical application of the performed studies on the operational formation of reservoir operation modes, since, being members of the interdepartmental working groups on the Volga-Kama and Angarsk reservoirs cascade, the authors use the developed methods and computational technologies in situational water resources management.



Paper [10] constructs a multi-objective optimal operation model in the upper reaches of the Yangtze River, integrating four objectives of power generation, ecology, water supply and shipping under the constraints of flood control. The OF for each target was determined. The constraints were: water balance constraint, reservoir discharge limits, reservoir water-level limits and power generation limits. For model solving, the non-dominated sorting genetic algorithm-III (NSGA-III) was used, which effectively prevents the falling into local optimum. In this study, 3 years were selected as a typical year for wet (1964), normal (1988) and dry (1959). The average monthly water level values in three typical years were taken as the decision variables, and the total number of decision variables was 240. The starting and ending regulation water levels were both set as the normal storage water level. Taking the reservoir system composed of 30 controlled reservoirs in the upper reaches of the Yangtze River as the research object, this paper showed that power generation is the main factor that restricts the other benefit functions of the reservoir, and is also restricted by them. During reservoir operation, it is more likely to sacrifice part of power generation to improve the satisfaction of other benefits, among which the competitive relationship with ecological objectives is the most obvious.

Unfortunately, in [10], the calculations were carried out for one characteristic year for three water content options: wet, normal, dry. This did not allow us to assess the system reliability over a long-term period of operation, during which the characteristic periods (wet, normal, dry) can last for several years (4–6). In addition, the calculation results were assessed by the value of the OF, in which different components (power generation, ecology, water supply and shipping) are poorly comparable in importance (benefit). Therefore, in the studies adduced, hydrological series of sufficient duration (44 years) were used. This made it possible to assess the reliability of the WRS by statistical criteria (reliability, resilience, vulnerability), which have clear economic and physical meaning. Furthermore, in contrast to [10], a comparative assessment of optimization and traditional management methods was made. Although the application of the NSGA-III multi-criteria optimization algorithm used in [10] is of certain interest.

Study [11] reported a review on applications of animal inspired EAs to reservoir operation optimization. Among them are ant colony optimization, particle swarm optimization, artificial bee colony, firefly algorithm, cuckoo search etc. According to the authors, all the animal inspired EAs outperformed traditional methods of reservoir optimization, such as nonlinear programming (NLP) and dynamic programming (DP). The comparison results revealed that constrained, discrete and randomized varieties of the animal inspired EAs outperformed unconstrained, continuous and deterministic varieties, respectively, because of larger feasible search space, better solution quality and shorter computational time.

Article [12] provides a heuristic method overview of the non-animal inspired EA applications for reservoir optimization. EAs do not mimic the biological traits and group strategies of animal (wild) species. In total, 14 suitable non-animal EAs have been identified, such as genetic algorithm (GA), simulated annealing (SA) and differential evolution (DE) algorithms. Comparisons were made of the used advisors' effectiveness in the studied literature. Research has shown that GA is the most commonly used algorithm, followed by DE. Non-animal advisors outperform classical reservoir optimization techniques (such as nonlinear programming and dynamic programming) due to faster convergence, variety of decision space and efficient objective function estimation. Reservoir operation optimization methods, according to [12], can be divided into two main groups: classical methods and evolutionary or metaheuristic algorithms (EA). Linear programming (LP), dynamic programming (DP), stochastic dynamic programming (SDP) and nonlinear programming (NLP) are classical methods that suffer from the "curse" of high dimensionality and slow convergence. Therefore, EAs (inspired by biological phenomena) have been developed, which have become widespread due to their search ability to find solutions close to global ones. The article [12] compares several single-purpose and multi-criteria algorithms for solving tank operation problems. GAs have been used to find optimal reservoir rule (DS) curves under climate change conditions, and have performed better than the current rule

curves. GA has been shown to be up to three times faster than DDDP, and is more efficient in determining optimal reservoir rule curves.

It should be noted that although we used GA to find the optimal configuration of release rules (dispatch schedules) [1–3], however, classical optimization methods give the best results, both in terms of computation time and the value of the objective function. In general, the authors, in contrast to [11,12], prefer to use nonlinear programming methods (the method of generalized reduced gradient), both implemented in standard programs such as Solver (Solver Excel) and developed by the authors to solve specific optimizing problems of reservoir cascades operation [13,14]. This is since OF and constraints, as a rule, are continuous, piecewise differentiable functions (unfortunately, strongly nonconvex), for which the conjugate gradient method is applicable (see Section 3 of this article). In addition, when the number of variables exceeds 1000 and the calculation of OF takes a lot of processor time, genetic algorithms perform much worse than nonlinear programming (NLP) methods, since they do not allow enough mutations and recombinations to produce a "good" solution, whereas NLP allows you to immediately determine the "best" movement to a local minimum.

The authors disagree with the thesis [12] "Optimization methods are designed to overcome the high dimensional, dynamic, non-linear, and stochastic features of reservoir systems". Optimization, according to the authors, solves three problems: it determines the reservoir basin capacity to meet the water users' requirements, allows to assess the quality of the adopted release rules (DS), justifies strategic planning measures to improve the performance of the WRS, and the fourth, presented in this article, allows the formation of the operating modes of the reservoir in real time. Although it was shown in [12] that non-animal inspired EAs were widely used to solve strategic planning problems and improve release rules by optimization methods, there are no publications on the use of optimization for the operational (in the real time) formation of reservoir operating modes. In [12], it was proposed to use optimization methods to assess the impact of climate change on the reservoir operation, and to search for effective algorithms for solving the problems of operation, which was completed by the authors of this article.

Unfortunately, the article [12] does not provide comparative estimates by statistical criteria of the optimization result's effectiveness for various algorithms. The quality of the used optimization method is assessed by the value of the OF, which sometimes poorly determines the actual benefit. There is no single test case (or several examples of different genesis) on which it would be possible to estimate the optimization time, the quality of the solution by the OF and the results of the analysis by statistical criteria (reliability, resilience, vulnerability) for different algorithms. Therefore, the recommendations made in [12] on the use of various methods are extremely dubious. We strongly recommend using statistical criteria to evaluate various optimization methods.

In this article (in contrast to [8–12]), the authors formulated a mathematical model, algorithm and computational technology, which allow, on the basis of optimization methods, to form in real time (for the current time interval) releases from the reservoir. To implement the optimization approach, a proprietary method was developed that allows the solving of large tasks in a reasonable time. A decomposition by time intervals is made, which makes it possible to significantly reduce the dimension and time of solving the optimization problem based on balance equations. Genetic and EA algorithms work well when the OF calculation time is short, otherwise the obtained local minimum may differ significantly from the global one. This article formulates the rules for the reservoir operational management, based on optimization methods, which, according to the authors, should more reliably fulfill the water users' requirements in comparison with traditional methods in the context of climate change.

In [15], a methodology for using single time step optimization (STO) and multiple time step optimization (MTO) solutions for river basin planning and real-time operation is presented. The results were obtained using a new water management model (web Basin Model WEB.BM) with a full set of linear programming (LP) optimization capabilities. The

variables to be optimized were releases from reservoirs that maximize OF and assume known forecasts of runoff and water demand. Several approaches to modeling are outlined: based on release rules (DS) and based on optimization. It is argued that there is no single favorite river basin management model widely used by practitioners. The capabilities and performance of a model can only be demonstrated by using it to provide successful solutions to complex test tasks. MTO is rarely used by practitioners due to: (a) few computing platforms can provide solutions in a reasonable amount of time; (b) debugging requires above average technical skills; (c) there are no reliable hydrological forecasts. In [15], it is shown that there is an effective way to develop and interpret MTO solutions that can help revise existing operating rules and use MTO as an operational tool for working with a reservoir in real time over a short time horizon, based on a combination of revised operating rules and short-term runoff forecast. The potential value of this approach is shown using the example of India. Most of the river basin planning and management models are based on DS. In fact, the best solution can only be obtained through optimal management of water demand (hedging), especially in dry years. The main advantages of MTO simulation are that there is no need to design a DS. Thus, statistical analysis of the model results can provide insight into the best operating rules, especially when they are based on a large number of modeled hydrological input series. The MTO approach is not entirely new, but until now there has been no accepted methodology for using long-term ideal model solutions to improve future reservoir operations in real time when inflow is unknown. The stochastic inflow prediction model is more reliable in estimating natural runoff than rain runoff models. They should only be used as a last resort to estimate natural runoff. If the operating rules outlined in [15] are followed, the model can achieve very close performance to the optimum one obtained using an ideal forecast for an entire hydrological year. The main conclusions formulated in [15]: river basin management, as a rule, is based on the use of DS which is poorly substantiated from a scientific point of view; the use of optimization models in combination with stochastic models can be very effective in constructing rules for the operation of a reservoir. This strategy can be combined with short-term runoff forecast and short-term optimization to manage future reservoir releases in real time.

In this article (in contrast to [15]), a statistical forecast is carried out for several years (tens of years) for the last years of the historical inflow, which reflect the ongoing climatic changes, and on this series the optimization task of the releases formation is solved, starting from a given initial reservoir volume. The optimization task is obtained with a dimension of about 1000 variables, which is solved by special methods developed by the authors in a classical deeply nonlinear setting. The centennial hydrological series of the observed inflow (and this is shown by the calculations) are not suitable for the formation of DS, since in this case, the average statistical runoff parameters are used, which differ significantly from the hydrological situation in recent years. The task solved in the present studies, including hydraulic calculations, cannot be linearized without losing its physical meaning, which is difficult to compensate for by further optimization. In addition, for the analysis of the calculation results, statistical efficiency criteria were used, which determine the reliability of the WRS and the quality of control, in contrast to [15], where the evaluation was carried out by the value of OF.

The review presents the most recent works, including the application of criteria for assessing the water resources quality management in a changing climate when solving various water resource problems using optimization methods. In [5–7], the results of the efficiency statistical criteria application are given; in [8–12,15], the use of optimization methods for the formation of the reservoir operating modes is shown. It should be noted that in all publications (except [15]), optimization is used only as a means of strategic planning and as a tool for building "good" release rules.

The studies have assessed climate changes in recent years and how these changes affect the management of the reservoir, according to the accepted release rules (DS). For this, two periods of 44 years of historical Inflow time series were taken: from 1932 to

1976, on which the release rules were based, and from 1976 to 2020, for which these rules are still applied. In addition, two 44-year series for 2020–2064 were modeled with the average parameters of the entire hydrological time series from 1903 to 2020 and a low-water series from 1995 to 2020. For these four rows, water resource calculations were carried out according to the current rules, and with the help of optimization with the modern requirements of water users. Calculations have shown the genesis of climate change, i.e., how climate change has impacted the ability to meet the requirements of water users. On the basis of multi-criteria analysis by statistical criteria, it was shown that the operating DS does not provide control with normative reliability in recent years and in the low-water period, but the optimization approach does. The analysis showed the following, it is necessary: (1) to develop DS on historical hydrological series over the past 20–30 years, and not on the entire series, which distorts the coordinates of the DS, (2) update DS every 10–15 years, (3) create tools that allow you to control in real time using an optimization approach. Such a mathematical model and algorithm were developed and proposed in the article. For its implementation, a long-term hydrological forecast is required.

Since the formation of the reservoir operation modes for the next period of time significantly depends on the forecast time series of inflow, it is necessary to conduct special research of the dependence of such a forecast series on the actually observed (historical) inflow series. Sections 2.1 and 3.1 describe a method for generating a forecast inflow hydrograph. Although the described methodology gives good results for a statistically justified forecast for any given time interval, however, for the next 10–20 years, such a forecast does not give satisfactory results. The variability of climate is controlled by natural processes, such as ocean cycles, changes in solar activity, volcanic activity and anthropogenic factors. The largest contribution comes from solar activity, which has been documented by a large body of published case studies. For a long-term multi-year forecast, the authors of the article recommend an approach based on the research outlined in the article by Laurenz et al. [16]. The methodology is based on the fact that the amount of precipitation in a particular region correlates quite well with the 11-year solar cycle of Schwabe. Although the authors did not find a significant 11-year dependence, according to the authors, this approach can allow the development of an algorithm for constructing a calculated long-term hydrological forecast of the inflow, and prove its effectiveness in comparison with the approach described in Section 2.1. However, this is a different manuscript.

## 2. Materials and Methods

In recent years, much attention has been paid to possible climate change problems. Attempts have been made to estimate these changes based on climate and runoff models. In the first part of the research, authors tried to assess climate change based on multi-criteria analysis method application using statistical performance criteria. For this, from the complete observed inflow hydrological series from 1903 to 2020, two 44-year series were identified from 1932 to 1976 and from 1976 to 2020, which corresponded to inflow used in operating release rule (DS) construction and current basin hydrological state. For these two series, water resource calculations were performed, defining releases for all calculated hydrological series intervals, according to DS and optimization methods. The calculation results were evaluated according to reliability, resilience and vulnerability criteria. In addition, two more hydrological series were constructed using stochastic methods: with average inflow indicators for the entire historical series and low-water period parameters, from 1995 to 2020. These two series were used as a possible forecast for 2020–2064. For these series, calculations were also performed and an analysis was carried out using DS and optimization methods.

The historical time series of the inflow from 1903 to 2020 were taken from the site of the Yenisei Basin Administration (http://pivr.enbvu.ru assessed on 24 October 2013), from the section "Rules for the use of water resources of the reservoirs water resources of the Angarsk HPP cascade (Irkutsk and Lake Baikal, Bratsk and Ust-Ilimsk)", file "Explanatory

note to the draft rules for the use of the Irkutsk reservoir water resources and Lake Baikal", "Natural average ten-day (V-IX) and average monthly water discharges, seasonal and annual volumes of useful inflow into the Lake Baikal for 1903/04/2012/13". Actual water user requirements for 2013 were taken from the same file. The inflow data have been updated until 2020 from the RusHydro website (http://www.rushydro.ru/hydrology/informer/ accessed on 11 September 2013).

When developing control models for large water bodies, a simplified equation of water balance is considered. As a characteristic of the management resource, a useful inflow was used, equal to the sum of the inflow of flowing rivers, the filtration inflow of groundwater, the sum of atmospheric precipitation entering the water surface minus evaporation. This simplification is due to the fact that the existing network of meteorological stations does not allow measuring rainfall and evaporation with the necessary error, while the useful inflow is calculated by inverse calculations from the water balance equation, and includes all poorly defined components of the water balance equation.

In the second part of a research, based on the results obtained by optimization methods, a mathematical optimization model, an algorithm and computational technique were developed that make it possible to form (without using DS) reservoir operating modes in real time (for the current calculated time interval, length from 10 to 30 days, depending on season) on the basis of predicted long-term hydrological inflow series, conflicting hierarchical order by prioritizing water users requirements (criteria) and reservoir initial volume (level) of the reservoir for the current calculated interval.

To solve the nonlinear programming optimization task, a special algorithm was developed that allows, due to interval decomposition, to significantly reduce the dimension and problem-solving time. At this point, the proposed computational technique is implemented in the Excel environment in VBA programming language using the Solver Excel optimizer. Figure 1 shows a flowchart of the developed computational technology.

### 2.1. Construction of a Forecast Hydrological Series

For a probabilistic forecast of inflow for a long period, it is necessary to adopt a hypothesis about the stochastic model of long-term runoff fluctuations. In modern conditions, there are violations of the stationary hydrological regime of rivers, which are largely due to climatic changes and are complex [17–19].

In the statistical processing of non-stationary sequences of this kind, the task of determining the points in the time series to which the process state changes arise. To characterize the properties of the state change point, it is necessary to consider this point as one of the parameters of the stochastic model in the non-stationary case, and apply one of the known estimation methods, for example Bayesian [20,21].

This setting of the task is known in the scientific literature as the task of finding the CPD (change point detection) change point. For the analysis of river runoff series in a non-stationary case, the approach developed in [20] was applied, based on the assumption that at certain moments there is a change in the state of a random process, and the observed sample is a set of homogeneous time sequences characterized by a random change of states. Next, consider the case where the parameters of probability models for homogeneous (stationary) sections are known.

We will consider a case following [20] when there is an accidental sequence $x_1$, $x_2$, $\cdots$ $x_n$, which is divided into two parts in r point ($1 \leq r \leq n$) and $x_i$ is distributed under law $F_1(x/\Theta_1)$, $i = 1, \cdots, r$, $x_i$ is distributed under law $F_2(x/\Theta_2)$, $i = r + 1, \cdots, n$ and $F_1(x/\Theta_1) \neq F_2(x/\Theta_2)$. The task is that on the set sequence of observations $x_1$, $x_n$, and the known functions of thr distribution of $F_1$ and $F_2$, it is necessary to draw a conclusion concerning a point of change of a condition of process.

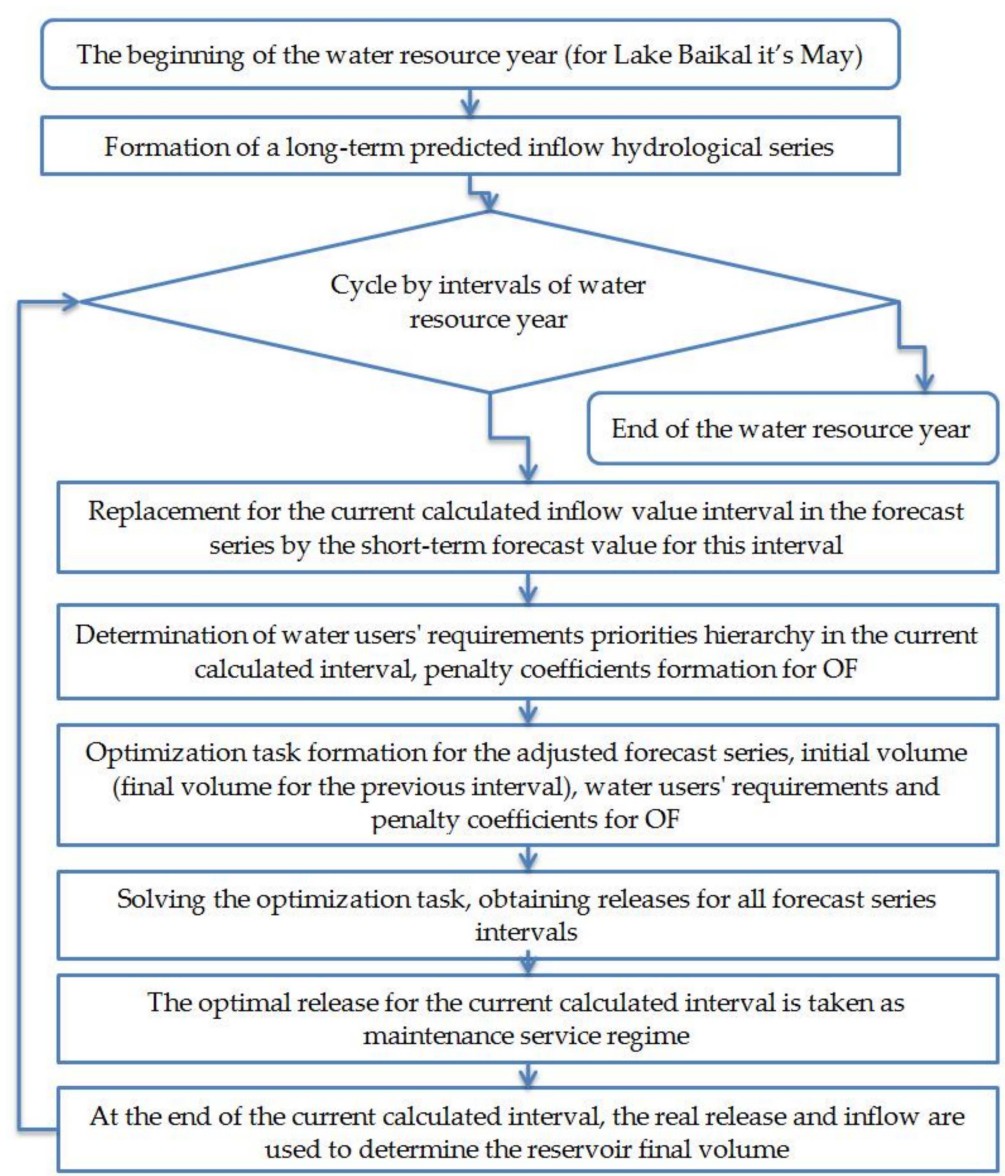

**Figure 1.** The flowchart of reservoir operating modes formation algorithm in real time.

Adopting log Pearson laws as density of distributions $\rho_1\ (x/\Theta_1)$ and $\rho_2\ (x/\Theta_2)$, joint distribution of sample $x_1, \cdots, x_n$ conditional in relation to parameters $\Theta_1$ and $\Theta_2$ and r having a point of change $(1 \leq r \leq n)$, we will write down:

$$
\rho\ (x_1, \cdots, x_n\ |\ r,\ \Theta_1,\ \Theta_2)\ =\ \rho_1\ (x_1, \cdots, x_r\ |\ \Theta_1) \cdot \rho_2\ (x_{r+1}, \cdots, x_r\ |\ \Theta_2)\ =
$$
$$
=\ \prod_{i=1}^{r} \rho_1(x_i\ |\ \Theta_1) \cdots \prod_{i=r+1}^{n} \rho_2(x_i\ |\ \Theta_2) \tag{1}
$$

A priori distribution r, having given $\rho0\ (r)$, $(1 \leq r \leq n)$, we define from the condition $\rho0\ (1) + \rho0\ (2) + \cdots + \rho0\ (n) = 1$.

According to Bayes's theorem, if $\Theta_1$ and $\Theta_2$, a posteriori distribution density of a point of change for the available observations $x_1, \cdots, x_n$, we will define from the general formula of total probability (Bayes formula):

$$
\rho(\ \Theta|x\ ) = \frac{\rho(x|\Theta) \cdot \rho(\Theta)}{\widetilde{\rho}(x)} \tag{2}
$$

where $\rho(x|\Theta)$—credibility of data x at a preset value of parameter $\Theta$:

$$\rho(x|\Theta) = \prod_{i=1}^{n} \rho_1(x_i|\Theta) \tag{3}$$

and $\widetilde{\rho}(\cdot)$ is calculated in the equation:

$$\widetilde{\rho}(x) = \int_{\Theta} P(x|\Theta) \cdot \rho(\Theta) d\Theta \tag{4}$$

and plays the role of a normalizing factor.

For the change point evaluation task, Equation (2) is written as:

$$\frac{P_n(r|\Theta_1, \Theta_2)}{P_0(r)} = \frac{\rho_1(x_1, \cdots, x_r|\Theta_1) \cdot \rho_2(x_{r+1}, \cdots, x_n|\Theta_2)}{\sum_r \rho_1(x_1, \cdots, x_r|\Theta_1) \cdot \rho_2(x_{r+1}, \cdots, x_n|\Theta_2) \cdot \rho_0(r) \cdot \frac{1}{n}} \tag{5}$$

Equations (1) and (5) are valid for sequences of independent and equally distributed random variables forming stationary sequences.

### 2.2. Water Resource Calculation Based on the Dispatch Schedule

In accordance with the regulatory documents of the Russian Federation (RF) [22] for the management rule development and reservoir operation reliability assessment, water resource, hydraulic and water-energy calculations are carried out for a long-term hydrological inflow series. Water resource calculations (WRC) make it possible to determine the reservoir operating modes (releases to the downstream and the levels of the upstream and downstream of the reservoir). WRC are carried out by solving the water balance equation for each calculated time interval.

The management quality is determined by water user's reliability on the number of years (intervals) without violations. A year is considered a failure if there are violations in at least one interval. As a management tool, a dispatch schedule (DS) is built, which sets releases from reservoir depending on the time interval and reservoir water level (volume) at the interval beginning (end). The DS coordinates are determined on the basis of simulation and optimization models that implement water resource calculation [2,23] and give acceptable (trade-offs) reliability parameters.

The permissible reliability range for each water use type is specified by regulatory documents [22]. The DS design process finishes if the reliability falls within the normative ranges. If this fails, then a compromise DS is built. Figure 2 shows the approval for the "Lake Baikal–Irkutsk Reservoir" DS 1988 [24] in graphical and tabular form, with a given releases range for each zone, depending on the levels and time interval.

Authors developed an algorithm and a computer simulation model in Excel written in the VBA language, which allows one to perform water resource calculation for a given hydrological inflow series and DS, and to determine for each water user (criterion) its annual and interval reliability, resilience and vulnerability.

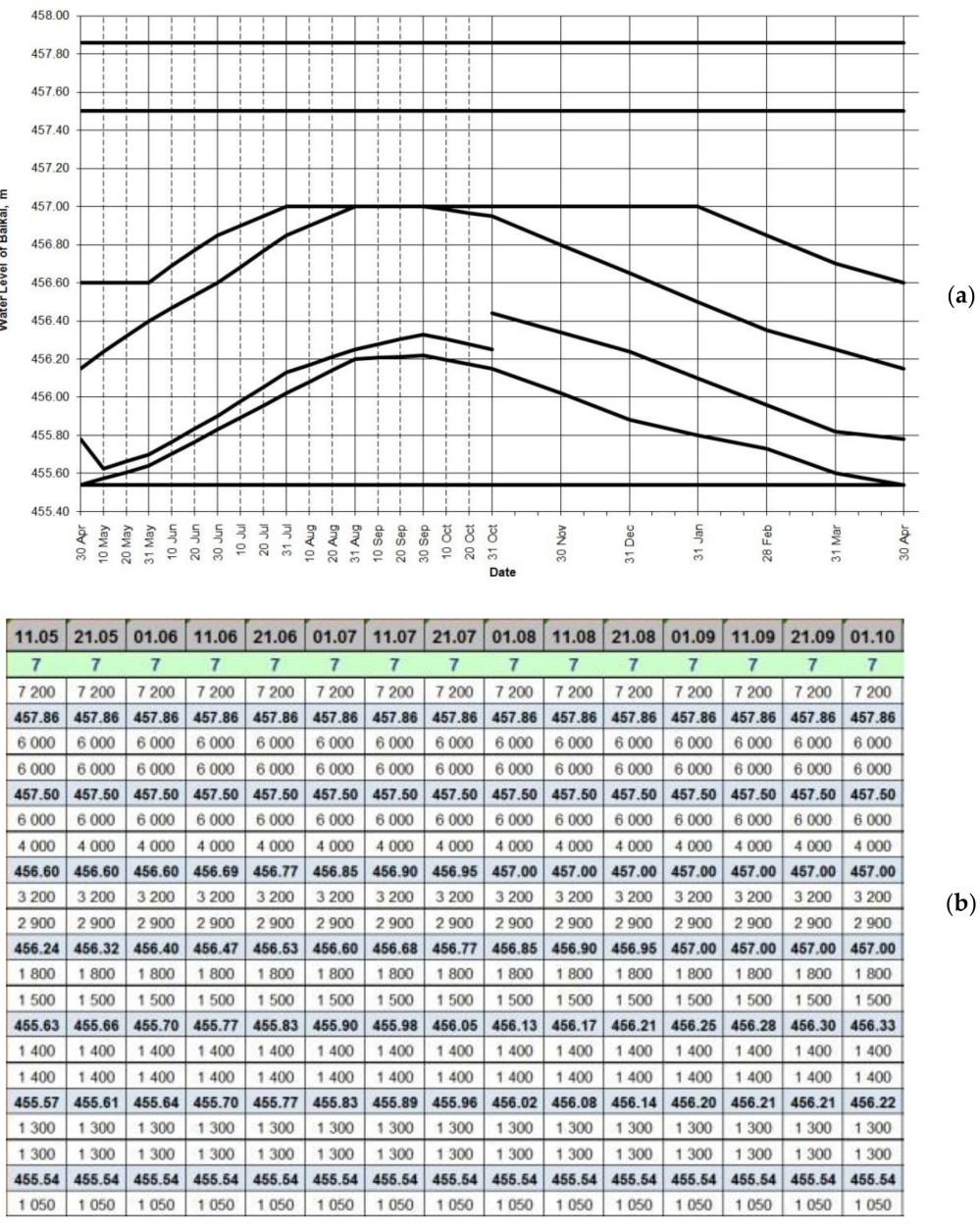

**Figure 2.** Shipment schedule for 1988 in graphic form (**a**) and fragment in tabular form (**b**).

*2.3. Water Resource Calculation by Optimization Methods*

Designing a dispatch schedule (DS) is a complex process. It is difficult to assess the DS management effectiveness. The DS management quality essentially depends on the priority hierarchy of water users' requirements, expressed in reliability criteria. If all water user requirements are satisfied with the normative reliability, then the DS can be put into operation (an acceptable decision on DS). However, if water users' requirements conflict and the WRC on DS does not give normative results in reliability terms, then it is necessary to find such a configuration of DS that increases the water users' reliability requirement values, depending on their priorities. At the same time, the DS developer always has three problems:

(1)  Are the DS parameters (coordinates) well chosen, i.e., a valid DS solution exists, but developer has not found it.

(2)   How "well" are water users' requirements being met in the adopted hierarchy? Is it possible to select the DS parameters so as to improve the reliability indicators for conflicting requirements?

(3)   Is the DS management optimal or is it possible to achieve better results using other management tools? Does the reservoir "water capacity" of the and catchment area allow it to satisfy the water users' requirements?

The first two problems are solved on the basis of methods for finding the optimal co-ordinates of the DS [2,23]. Although a complex, nonlinear, discrete (piecewise-continuous) optimization problem of a small dimension (in our case, 168 independent variables) arises, its solution requires great skill and special optimization methods (the authors used special methods of local optimization and genetic algorithms). These methods do not give a global optimum, but they significantly improve the reliability indicators of water supply to users.

The third problem is related to the solution of a rather complex optimization problem of a large dimension (in our case, $24 \times 44 = 1056$ independent variables), which requires the development of special optimization methods, since it is almost impossible to solve an optimization task of such a dimension in a reasonable time using nonlinear programming (NLP) methods. However, only its solution gives (or refutes) confidence in the quality dispatch management. Below is the author's algorithm for solving the third problem.

The meaningful formulation of the optimization task is as follows: it is necessary to determine the releases (independent variables) from the reservoir for the entire calculation period, which minimize a certain objective function (OF), under constraints associated with the fulfillment of balance equations (reservoir volume r in the next time interval is equal to reservoir volume in the current interval plus inflow minus release). The OF is calculated depending on water users' requirements and should decrease with the decrease in failure events (when required releases violate the threshold). Water users' requirements priorities hierarchy is reflected in the OF by means of penalty coefficients (the higher priority, the higher penalty for requirements violation). The initial data for the optimization task are the interval inflow series, initial and final reservoir volumes.

### 2.3.1. Mathematical Model and Algorithm for Solving the Optimization Task

The mathematical model of finding optimal releases to the Irkutsk reservoir downstream task for a given long-term inflows series is based on solving a system of interval balance equations:

$$W_{i+1} = W_i + P_i - R_i \tag{6}$$

Here $i$ is the time interval, $W_i$ is the water volume in Lake Baikal at the interval beginning $i$, $P_i$ is the volume (or average discharge) of "net" inflow (without losses for evaporation and seepage) to Lake Baikal in the interval $i$, $R_i$ is the volume (or average discharge) of release to the Irkutsk reservoir downstream in interval $i$.

The main dependent variables of the WRC are calculated using the known functions for the "Lake Baikal–Irkutsk reservoir" complex:

$$Z_i = F_B(W_i)\text{—bathymetric dependence function of the Lake Baikal level } \mathbf{Z_i} \text{ on the Lake Baikal volume;} \tag{7}$$

$$H^u{}_i = F_u(Z_i, R_i)\text{—dependence function of the Irkutsk reservoir upstream level } \mathbf{H^u}{}_i \text{ on the lake Baikal level and releases into the Irkutsk reservoir downstream;} \tag{8}$$

$$H^d{}_i = F_d(R_i)\text{—dependence function of downstream level } \mathbf{H^d}{}_i \text{ on the Irkutsk reservoir release into the downstream} \tag{9}$$

$$N_i = N(H^u{}_i, H^d{}_i, R_i)\text{—hydroelectric power dependence function on headwater level, tailwater and release into the Irkutsk reservoir downstream} \text{ (difference on the head—} H^u{}_i - H^d{}_i) \tag{10}$$

Water users' requirements form 12 criteria:

1. The Lake Baikal level should be in range (455.54, 457.5) m;
2. The Lake Baikal level should be $\geq$456 m, [25];
3. The Lake Baikal level should be $\leq$457 m, [25];
4. The maximum release in winter should be less than 2500 m$^3$/s;
5. The transport release during navigation should be more than 1500 m$^3$/s;
6. The release for water supply should be in range of (1250, 1300) m$^3$/s;
7. Flood control release should be less than 3200 m$^3$/s;
8. Guaranteed winter power should be more than 347 MW;
9. The Irkutsk reservoir upstream level for water intakes operation should be more than 454 m;
10. The pressure on the dam for HPP operation should be more than 26 m;
11. The Lake Baikal level on May 1 for normal fish spawning should be 456.15 m;
12. The Lake Baikal level during September for normal fish spawning should be 457 m.

The water users' requirements priorities hierarchy in modern conditions is as follows: (1) municipal water supply, sanitary release criterion 6, 9; (2) transport 5; (3) fisheries, 11, 12; (4) flooding of the area, 1, 4, 7; (5) energy, 8, 10; (6) decree no. 234 [25], 1, 2, 3. At the time of DS of the 1988 development, the priorities were as follows: (1) transport, 5; (2) energy, 8, 10; (3) flooding of the area, 6000 m$^3$/s (criterion 7); (4) fisheries, 11, 12.

It can be seen from the above data that the priorities have changed significantly. In addition, after the Irkutsk floodplain development, the maximum release is 3200 m$^3$/s.

The optimization approach used when performing water resource calculations (WRC) is to find the "best" releases into the Irkutsk reservoir downstream (independent variables), ensuring the reliability of water users' hierarchically ordered requirements [13,25]. The calculations were performed for a given long-term average inflow hydrological series over the time interval, and the initial (in the starting series interval) and final (in the ending series interval) reservoir volume.

Figure 3 shows the optimal solution algorithm flowchart findings for the given optimization task.

The mathematical task can be formulated as follows. Consider water balance equation system for a reservoir with a given long-term inflow series containing **T** years and **M** intervals per year (in the article **T** = 44, **M** = 24):

$$W_{11} + P_{11} - R_{11} = W_{12}, \ldots, W_{1m} + P_{1m} - R_{1m} = W_{1m+1}, \ldots, W_{1M} + P_{1M} - R_{1M} = W_{21},$$
$$\ldots \ldots \ldots \ldots \ldots \ldots \ldots \ldots$$
$$W_{t1} + P_{t1} - R_{t1} = W_{t2}, \ldots, W_{tm} + P_{tm} - R_{tm} = W_{tm+1}, \ldots, W_{tM} + P_{tM} - R_{tM} = W_{t+11}, \quad (11)$$
$$\ldots \ldots \ldots \ldots \ldots \ldots \ldots \ldots$$
$$W_{T1} + P_{T1} - R_{T1} = W_{T2}, \ldots, W_{Tm} + P_{Tm} - R_{Tm} = W_{Tm+1}, \ldots, W_{TM} + P_{TM} - R_{TM} = W_{T+11}$$

Here $t$ is the year number, $m$ is the interval number of the long-term hydrological series, $t = [\mathbf{1,T}]$; $m = [\mathbf{1,M}]$; $\mathbf{W}_{tm}$ and $\mathbf{W}_{tm+1}$ are the initial and final reservoir water volumes; $\mathbf{W_{11}}$, $\mathbf{W_{T+11}}$ specify starting and ending volumes. In accordance with the normative document [22], $\mathbf{W_{11} = W_{T+11}}$ is accepted in Russia. This provides an overall water balance for the entire system (11).

It is required to find the optimal release values $\mathbf{R}_{tm}$ for the given inflow values $\mathbf{P}_{tm}$, fixed $\mathbf{W_{11}}$, $\mathbf{W_{T+11}}$, at which all water users' requirements will be satisfied as much as possible. The number of intervals (or years) in which water users' requirements (criteria) are violated is minimized in accordance with their priorities. Furthermore, in accordance with the standard [22], it is necessary to minimize the violation depth, the difference between the threshold and the criterion value, in case of failure. In this case, the optimization task becomes multi-criteria.

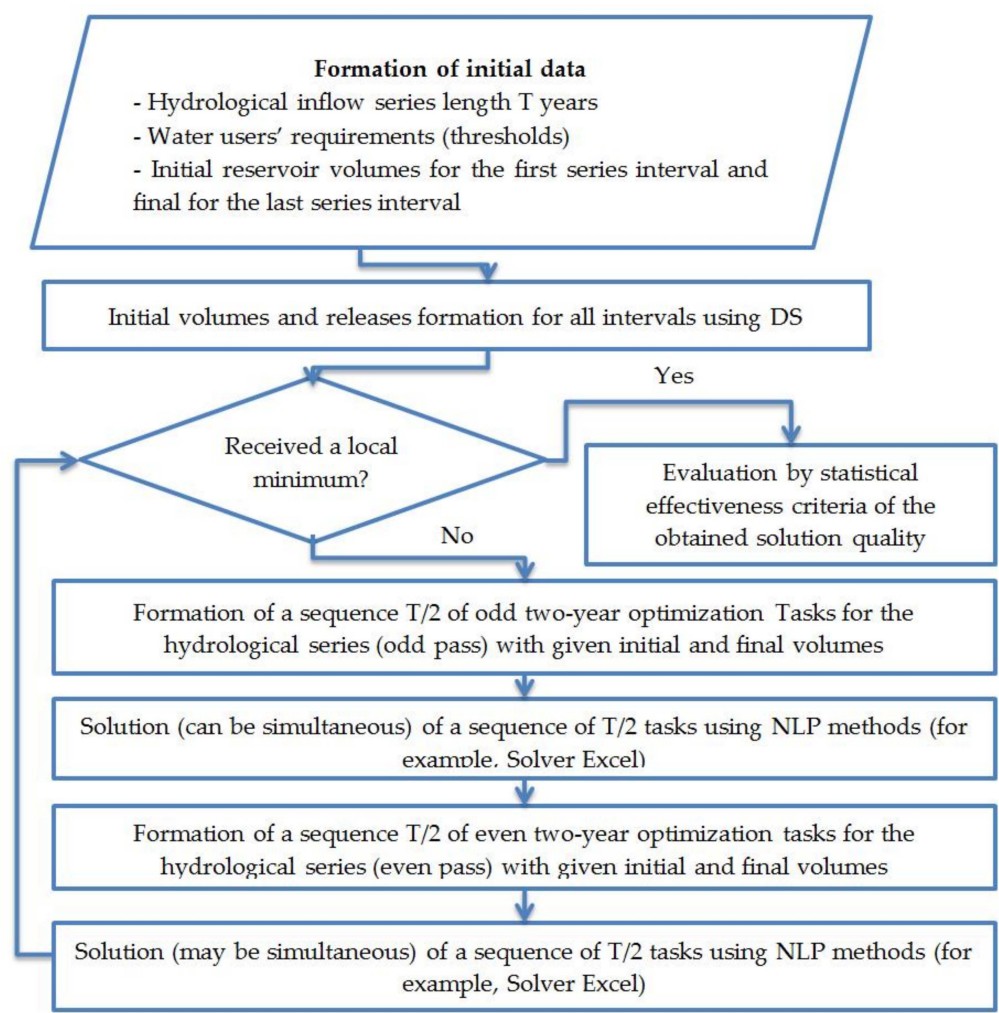

**Figure 3.** The algorithm flowchart of optimal releases for solving a reservoir task.

The integer nonlinear optimization task of such a dimension (the long-term flow series used in the article generates 1056 independent variable releases in the optimization task) with a large number of local minima cannot be solved in a reasonable time by a standard program, such as a standalone Solver (especially MS Excel Solver), due to "Curse of Dimension". Therefore, the authors proposed their own version of an optimization algorithm, called by the authors as the "Pulsating (oscillating) Spring method". Moreover, the objective functions that determine the additive function were represented by the square requirement (threshold) deviation from the criterion value in the case of violation. This simplification reduces the number of violations in years with medium inflows, but may increase the number of violations in dry years. However, with this approach, the depth of the violation decreases (with a large even degree, the task turns into a minimax one, that is, the maximum depth becomes minimum), which is sometimes a more important indicator. This approach makes it possible to exclude integerness and remain in the class of continuous, piecewise differentiable functions.

An arbitrary number $K$ of criteria is allowed, each of which is given in the form of system control parameter functions defined through the variables $\mathbf{R}_{tm}$ and $\mathbf{W}_{tm}$. These functions are reservoir volumes and water levels, levels of up- and downstream, releases, hydropower and electricity generation and others. For example, the criterion may be the square of release deviation $\mathbf{R}_{tm}$ from allowable maximum $\mathbf{R}_{max}$ that does not lead to downstream flooding: if $\mathbf{R}_{tm} > \mathbf{R}_{max}$ then $(\mathbf{R}_{max} - \mathbf{R}_{tm})^2$, else **0.**

To account the water users' requirements in the regulation process, at each hydrological series interval, we define the square $\Delta^2{}_{tmk}(\mathbf{W}_{tm}, \mathbf{R}_{tm}, \mathbf{W}_{tm+1})$ of deviation of $k$th criterion

threshold in $i$th interval from the regulation result. In accordance with the significance of each requirement $k$, a penalty factor $\mathbf{C}_k$ is introduced. This is a weighting factor that determines the place of the requirement in the priorities hierarchy.

Let's denote $\mathbf{W} = \{\mathbf{W_{11}}, \ldots , \mathbf{W_{TM}}\}$, $\mathbf{R} = \{\mathbf{R_{11}}, \ldots , \mathbf{R_{TM}}\}$. The regulation quality is estimated using the objective function $\mathbf{F(W,R)}$ defined as the sum of squared deviations from water users' requirements for all intervals and all the criteria $\Delta^2{}_{tmk}(\mathbf{W}_{tm},\mathbf{R}_{tm},\mathbf{W}_{tm+1})$, taking into account the penalty factors:

$$F(W,R) = \Sigma_{k=[1,K]}\, C_k \times (\Sigma_{t=[1,T]}\, \Sigma_{m=[1,M]}\, \Delta^2{}_{tmk}(W_{tm},R_{tm},W_{tm+1})) \tag{12}$$

Note that this function is additive with respect to $\mathbf{K}$, $\mathbf{T}$ and $\mathbf{M}$.

The optimization task can be formulated as follows: to minimize functional (12) under constraints (11) and constraints for all variables $\mathbf{W}_{tm}$ and $\mathbf{R}_{tm}$:

$$W \geq 0, R \geq 0 \tag{13}$$

Below we describe an algorithm for solving the optimization task.

At the first stage, water management calculations are carried out for a long-term series inflow using a dispatch schedule that determines the amount of release depending on upstream levels and the time interval. The release and volume values obtained as a result of these calculations are used as initial values in the iterative process:

$$R^{(0)} = \{R^{(0)}{}_{11}, \ldots , R^{(0)}{}_{TM}\},\ W^{(0)} = \{W^{(0)}{}_{11}, \ldots , W^{(0)}{}_{TM}\}$$

The objective function (OF), in this case has some value $\mathbf{F^{(0)}(W^{(0)}, R^{(0)})}$.

Suppose, for simplicity, that the number of years is even (for an odd number of years, the algorithm is inverted). At the second stage (odd pass), the volume values in the first and last interval of an odd two-year (1, 3, 5, etc.) are fixed, and Solver.xlam for Microsoft Excel (Solver-Excel) solves the task of minimizing the objective function of the corresponding two-year, depending on releases. Thus, the problem of a large dimension (11)–(13) with dimension T*M is decomposed into T/2 independent subtasks of dimension 2*M, each of which is solved by Solver-Excel on average in 10 min. The mathematical setting of the $t$th two-year task: minimize objective function (OF):

$$F_{2t-1,2t}(W,R) = \Sigma_{k=[1,K]}\, C_k \times (\Sigma_{m=[1,M]}\, \Delta^2{}_{2t-1mk}(W_{2t-1m},R_{2t-1m},W_{2t-1m+1}) + {}$$
$$+\Sigma_{m=[1,M]}\, \Delta^2{}_{2tmk}(W_{2tm},R_{2tm},W_{2tm+1})) \tag{14}$$

Under the following constraints on the variables $\mathbf{W}$ and $\mathbf{R}$:

$$W_{2t\text{-}11} + P_{2t\text{-}11} - R_{2t\text{-}11} = W_{2t\text{-}12};$$
$$\ldots\ \ldots\ \ldots\ \ldots\ \ldots\ \ldots\ \ldots\ \ldots$$
$$W_{2t\text{-}1m} + P_{2t\text{-}1m} - R_{2t\text{-}1m} = W_{2t\text{-}1m+1};$$
$$\ldots\ \ldots\ \ldots\ \ldots\ \ldots\ \ldots\ \ldots$$
$$W_{2t\text{-}1M} + P_{2t\text{-}1M} - R_{2t\text{-}1M} = W_{2t\ 1};$$
$$W_{2t1} + P_{2t1} - R_{2t1} = W_{2t2};$$
$$\ldots\ \ldots\ \ldots\ \ldots\ \ldots\ \ldots\ \ldots \tag{15}$$
$$W_{2tm} + P_{2tm} - R_{2tm} = W_{2tm+1};$$
$$\ldots\ \ldots\ \ldots\ \ldots\ \ldots\ \ldots\ \ldots$$
$$W_{2tM} + P_{2tM} - R_{2tM} = W_{2t+1\ 1};$$
$$W_{2t\text{-}11} = W^{(0)}{}_{2t\text{-}11};\ W_{2t+11} = W^{(0)}{}_{2t+11};$$

$$W_{2t\text{-}1m} \geq 0, W_{2tm} \geq 0, R_{2t\text{-}1m} \geq 0\ R_{2tm} \geq 0, \forall m = [1,M] \tag{16}$$

Thus, the original $\mathbf{T*M}$ dimensional optimization task (11)–(13) splits into $\mathbf{T/2}$ (for even $\mathbf{T}$ and $\mathbf{(T-1)/2}$ for odd) subtasks of type (14)–(16), which can be solved independently from each other, hence one can use the multitask and multiprocessor computer modes. Such a decomposition of the optimization task into T/2 independently solvable subtasks allows

using the multitasking mode of the operating system (Windows) and the multiprocessor mode of supercomputer, which can significantly reduce the optimization time (practically by several tens of times). These properties are implemented for the high-level programming language C$^{++}$ of a multiprocessor supercomputer, on which the next version of presented computing technology is supposed to be implemented.

Let $R^{(1)}_{2t-1m}$, $W^{(1)}_{2t-1m}$, $R^{(1)}_{2tm}$, $W^{(1)}_{2tm}$, **m = [1,M]** be the variable values that realize the minimum of the OF (14). The $F_{2t-1,2t}(W,R)$ value does not increase compared to the approximation zero variable values $R^{(0)}_{2t-1m}$, $W^{(0)}_{2t-1m}$, $R^{(0)}_{2tm}$, $W^{(0)}_{2t1m}$, **m = [1,M]**. It follows that after solving all local subtasks (14)–(16), *t* = **[1,T/2]**, the value $F^{(1)}(W^{(0,1)},R^{(1)})$ of the OF (12) for the found vectors, $W^{(0,1)} = \{W^{(0)}_{11}, \ldots, W^{(0)}_{2t-11}, W^{(1)}_{2t-12}, \ldots, W^{(1)}_{2t-1m}, \ldots, W^{(1)}_{2t1}, \ldots, W^{(1)}_{2tm}, \ldots, W^{(0)}_{2t+11}, \ldots, W^{(1)}_{TM}, \ldots, W^{(0)}_{T+11}\}$; $R^{(1)} = \{R^{(1)}_{11}, \ldots, R^{(1)}_{TM}\}$, *t* = **[1,T/2]**, **m = [1,M]**, at least, will not increase compared to $F^{(0)}(W^{(0)},Q^{(0)})$.

At the third stage, all the found even volumes $W^{(1)}_{2t1}$, *t* = **[1,T/2 − 1]** (even pass) are fixed and the odd ones become variables. The decomposition procedure is repeated, but with fixed even volumes.

$$F_{2t,2t+1}(W,R) = \Sigma_{k=[1,K]} C_k \times (\Sigma_{m=[1,M]} \Delta^2_{2tmk}(W_{2tm},R_{2tm},W_{2tm+1}) + \Sigma_{m=[1,M]} \Delta^2_{2t+1mk}(W_{2t+1m},R_{2t+1m},W_{2t+1m+1})) \tag{17}$$

Under the following constraints on the variables **W** and **R**

$$W_{2t1} + P_{2t1} - R_{2t1} = W_{2t2};$$
$$\ldots \ldots \ldots \ldots \ldots \ldots \ldots$$
$$W_{2tm} + P_{2tm} - R_{2tm} = W_{2tm+1};$$
$$\ldots \ldots \ldots \ldots \ldots \ldots \ldots$$
$$W_{2tM} + P_{2tM} - R_{2tM} = W_{2t+11};$$
$$W_{2t+11} + P_{2t+11} - R_{2t+11} = W_{2t+12}; \tag{18}$$
$$\ldots \ldots \ldots \ldots \ldots \ldots \ldots$$
$$W_{2t+1m} + P_{2t+1m} - R_{2t+1m} = W_{2t+1m+1};$$
$$\ldots \ldots \ldots \ldots \ldots \ldots \ldots$$
$$W_{2t+1M} + P_{2t+1M} - R_{2t+1M} = W_{2t+21};$$

$$W_{2t1} = W^{(1)}_{2t1}; W_{2t+21} = W^{(1)}_{2t+21};$$
$$W_{2tm} \geq 0, W_{2t+1m} \geq 0, R_{2tm} \geq 0 \; R_{2t+1m} \geq 0, \forall m = [1,M] \tag{19}$$

Let $R^{(2)}_{2tm}$, $W^{(2)}_{2t+1m}$, $R^{(2)}_{2tm}$, $W^{(2)}_{2t+1m}$, **m = [1,M]** be the variable values that realize the minimum of OF (17). The $F_{2t,2t+1}(W,R)$ value does not increase compared to the value for the previous pass variables $R^{(1)}_{2tm}$, $W^{(1)}_{2t+1m}$, $R^{(1)}_{2tm}$, $W^{(1)}_{2t+1m}$, **m = [1,M]**. It follows that after solving all local subtasks (17)–(19), *t* = **[1,T/2−1]**, the value $F^{(2)}(W^{(1,2)},R^{(2)})$ of the OF (12) for the found vectors, $W^{(0,1)} = \{W^{(1)}_{11}, \ldots, W^{(1)}_{2t-11}, W^{(2}_{2t-12}, \ldots, W^{(2)}_{2t-1m}, \ldots, W^{(2)}_{2t1}, \ldots, W^{(2)}_{2tm}, \ldots, W^{(1)}_{2t+11}, \ldots, W^{(2)}_{TM}, \ldots, W^{(1)}_{T+11}\}$; $R^{(2)} = \{R^{(2)}_{11}, \ldots, R^{(2)}_{TM}\}$, *t*= **[1,T/2]**, **m = [1,M]**, at least, will not increase compared to $F^{(0)}(W^{(0)},Q^{(0)})$.

The process of alternating even and odd passes leads to finding a local minimum. The local minimum value, as in all similar tasks, can substantially depend on the initial solution. The cycle of even and odd passes is repeated several times until the function stops decreasing with the accepted accuracy. The process convergence to a local minimum is ensured by a monotonic decrease in the objective function (OF). Studies have shown that the solution choice obtained using the DS as the initial one significantly improves the convergence and decreases the OF value (the DS gives a good solution). In our calculations, it was enough to perform four cycles for convergence. The duration of the calculation of one cycle for a 44-year series was from 4 to 6 h with a conventional computer operation.

A procedure that implements the above-described algorithm for decomposing a multidimensional problem into a number of problems of lower dimension and sequentially solves local problems was developed in Excel using the Visual Basic (VBA) programming language for the "Lake Baikal–Irkutsk Reservoir". The procedure uses a system of water balance Equation (13) with the indicated 12 criteria.

As an example of convergence, Figure 4 shows a step-by-step change in the OF as a result of calculating the fourth cycles, each of which consisted of two passes: an odd one, including 22 odd steps, and an even one, including 21 even steps.

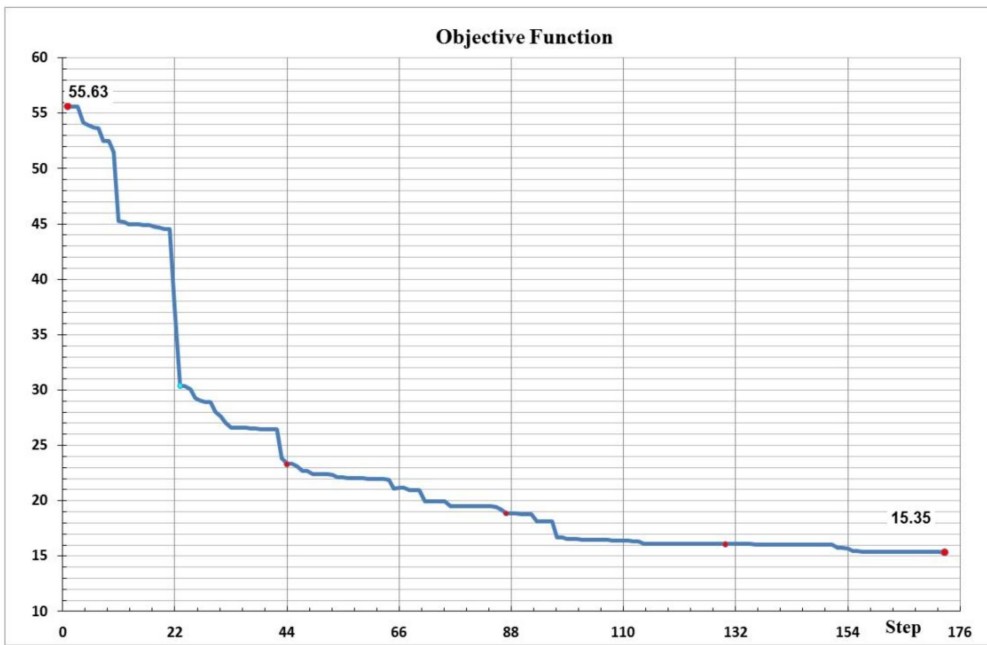

**Figure 4.** Dynamics of step-by-step changes of the objective function.

### 2.4. The Multi-Criteria Analysis Based on Performance Statistical Criteria

There are many ways to analyze stochastic time series data. The average value usually gives too little information on the genesis of long-term hydrological inflow series. It is very difficult to compare the time series themselves, both the initial data and calculation results. Let us define the statistical performance criteria used in the section for analysis. Let $X = \{X_t\}$ the time series data that is obtained as a result of water resource calculations (WRC) and $X^T$ is the threshold value for this criterion [4, Chapter 9].

Time series *reliability* can be defined as the number of data in a satisfactory state, divided by the total number of data in the time series. If satisfactory values in a time series $X_t$ containing $N$ values ($t$, time intervals) are equal to or exceed some threshold $X^T$, then:

$$\text{reliability}[X] = [\text{The number of time periods } t \text{ such that } X_t \geq X^T]/N \quad (20)$$

*or the* time series *annual reliability* can be defined as the number of years in a satisfactory state divided by the total number of years in the time series. If $N_{year}$ the total number of years, then:

$$\text{reliability\_A}[X] = [\text{The number of years in which for all periods t within a year,} \quad X_t \geq \geq X^T]/N_{year} \quad (21)$$

*Resilience* is defined as the satisfactory event occurrence probability in the time interval $t + 1$ after a failure event in the interval $t$. The resilience is calculated as follows:

$$\text{resilience}[X] = [\text{number of times a satisfactory value follows an unsatisfactory value}]/[\text{number of times an unsatisfactory value occurred}] \quad (22)$$

*Vulnerability* is defined as the average deviation failure time series values $X_t$ from a threshold value $X^T$ (average "depth" of failures). The vulnerability is calculated as follows:

$$\text{vulnerability}[X] = [\text{sum of positive values of } (X^T - X_t)] / \tag{23}$$
$$[\text{number of failures occurred}]$$

*Or the normalized vulnerability* is defined as the average deviation fraction from the maximum deviation (maximum "depth" of failures). The normalized vulnerability is calculated as follows:

$$\text{vulnerability\_N}[X] = [\text{sum of positive values of } (X^T - Xt) / \text{Max}\{(X^T - X_t) \mid t\text{-failures}\}] / / [\text{number of failure times occurred}] \tag{24}$$

## 3. Results

### 3.1. The Result of Predicted Hydrological Series Construction

The Bayesian method (see Section 2.1) of estimating the change point is applicable to the sequence of annual inflows to Lake Baikal (Figure 5a). Since 1995, it is believed that the runoff regime has changed, and we discuss two states of the hydrological system, characterized by different parameters. The change probability distribution point density calculation results show the presence of such an event in the hydrological series (Figure 5b).

In this case, the time series section, starting in 1995, can serve as a forecast model in the case of the heaviest scenario. Parameters defined across a range of observations will determine the optimal scenario.

Two water management scenarios are used to investigate the water management system. Scenarios are artificial lake inflow time series, obtained through simulations. To model artificial series, a probabilistic model in the Markov Chain form and a set of distribution parameters, obtained as a result of processing homogeneous parts of a multi-year inflow series, are used.

The homogeneous sections of the time series (Figure 5) are scenarios characterizing the system operation in normal and especially low-water conditions. Using the above Bayesian method, two 44-year forecast hydrological series were generated. The forecast for the period up to 2064 is represented by two scenarios: average and low water.

Figure 6 shows diagrams of two observed inflow series, 1932–1976 and 1976–2020, and two forecast series, 2020–2064 average and 2020–2064 low water, formed according to the average parameters of the entire series 1903–2020, and low-water parameters of the series 1995–2020. These series were used later for water resource management calculations. In Figures 6–10, the abscissa shows the number of the year in order from the range of years in the legend.

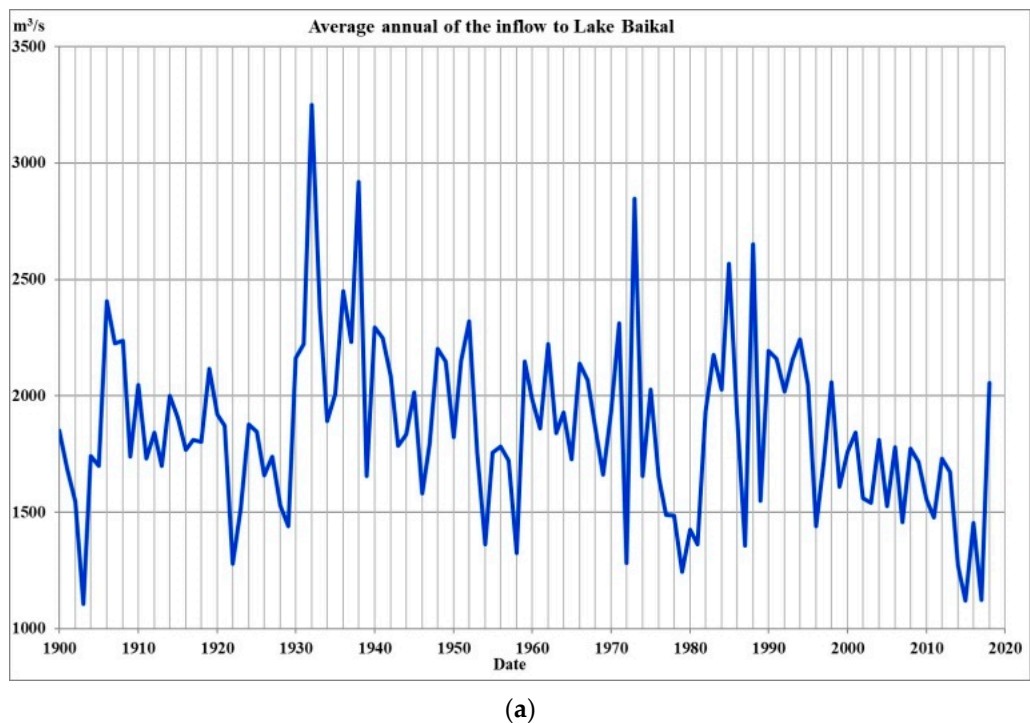

(**a**)

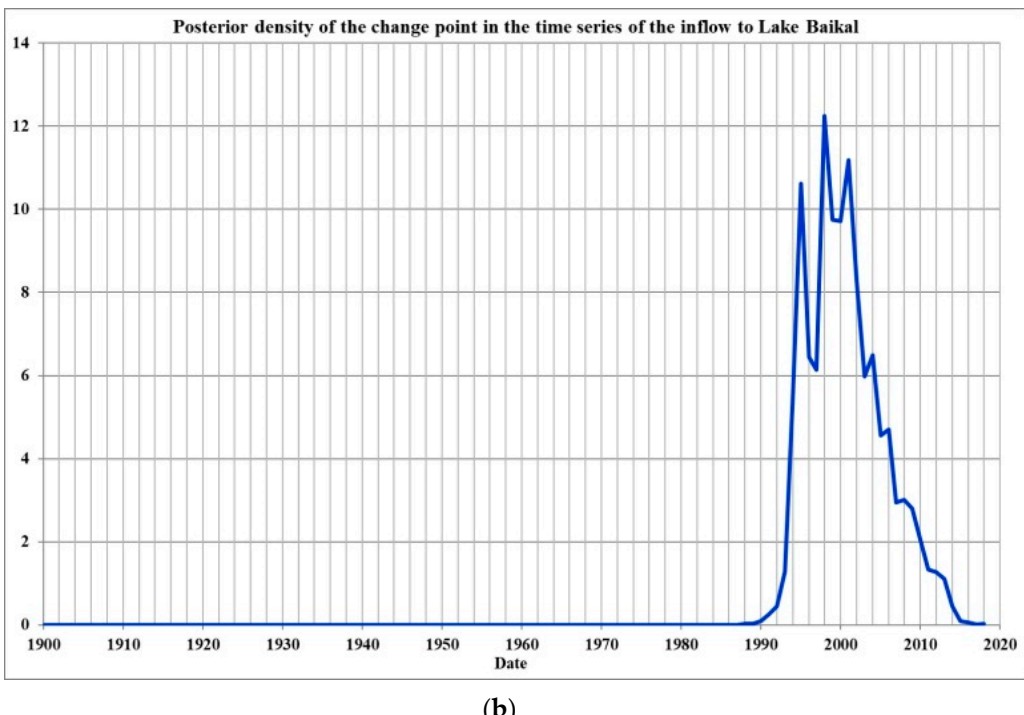

(**b**)

**Figure 5.** Chronological graph of the inflow average annual values of the Lake Baikal (**a**) and posterior density (Bayesian estimate) of the change point in the inflow time series to Lake Baikal (**b**).

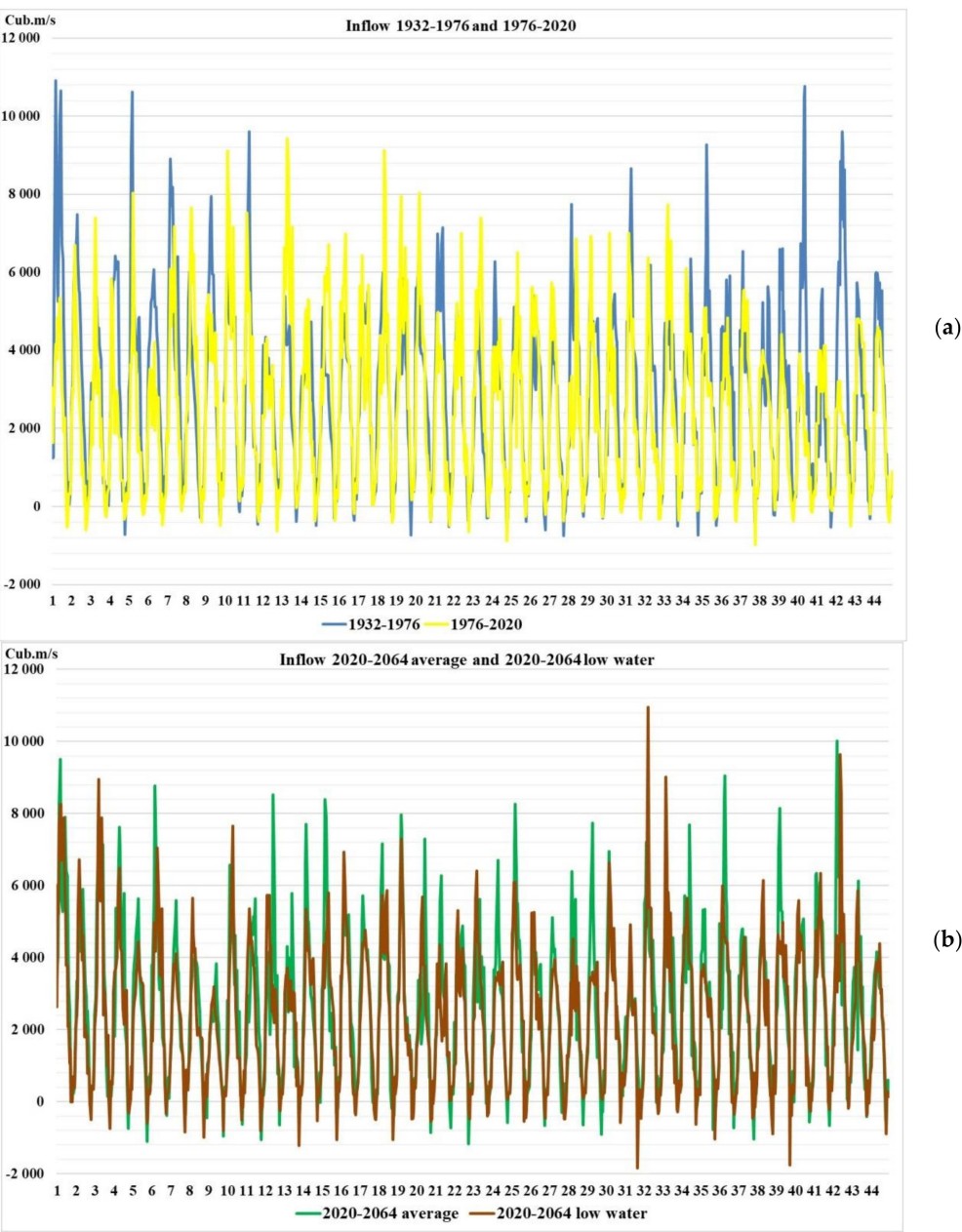

**Figure 6.** Long-term series of observed and forecast inflows. (**a**) 1932–1976 and 1976–2020; (**b**) 2020–2064.

### 3.2. The Water Resource Calculations Results Using Dispatch Schedule

Figures 7 and 8 show the results of WRC (water levels of Lake Baikal and releases in the Irkutsk reservoir downstream) for the DS 1988 for four hydrological series (two observed and two forecast). The calculations were carried out for the initial volume of Lake Baikal 456.15 m (the fishery threshold).

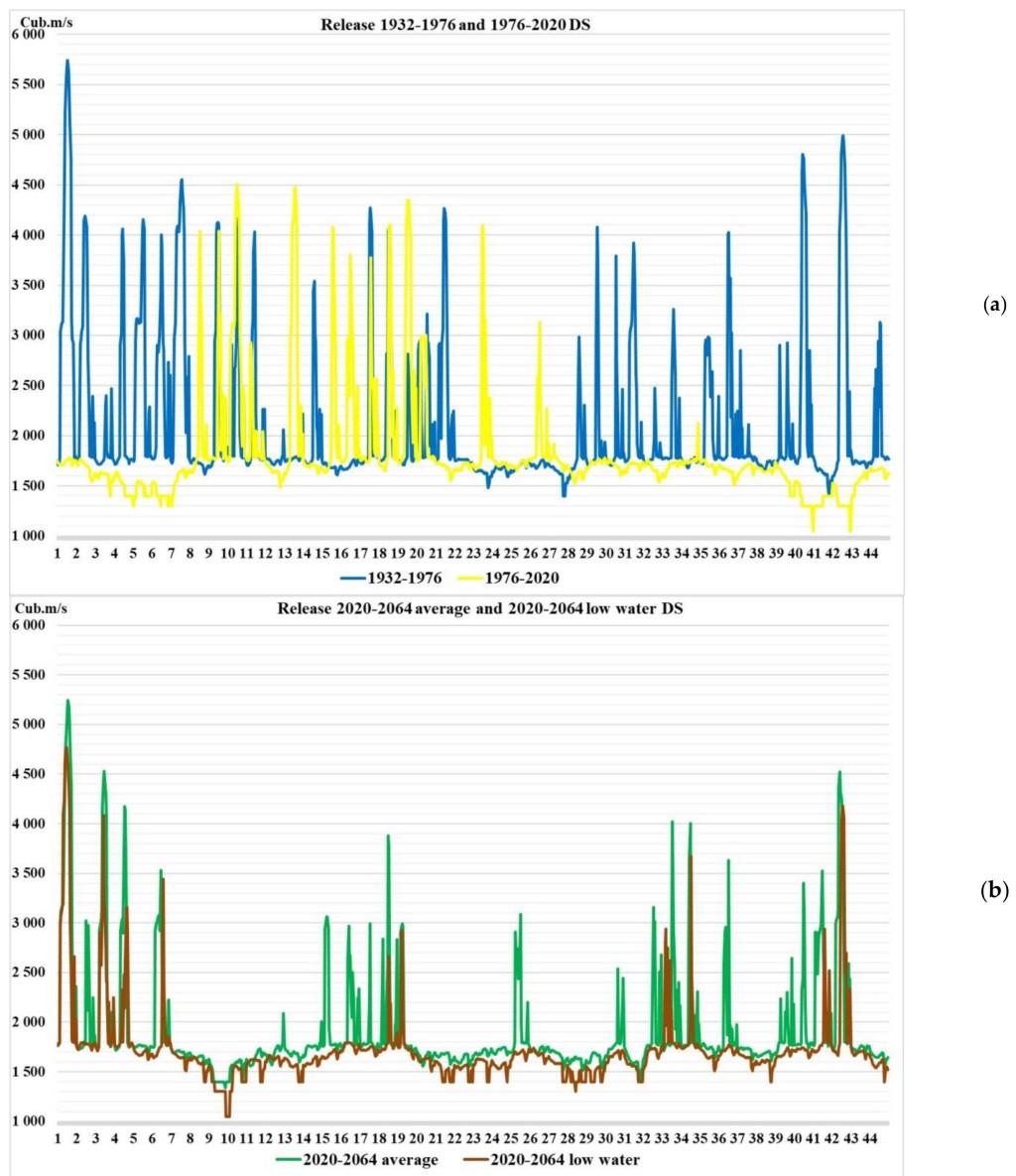

**Figure 7.** Releases to the Irkutsk reservoir downstream obtained as a result of WRC by DS for series 1932–1976 and 1976–2020 (**a**) for series 2020–2064 average and 2020–2064 low water (**b**).

*3.3. The Water Resource Calculation Result Based on the Optimization Methods*

For the four hydrological inflow series to Lake Baikal listed in Section 3.1 (Figure 6), water resource calculations (WRC) were performed based on the optimization methods given in Section 2.3.1. The calculations were carried out for the same initial volume of Lake Baikal (456.15 m is the fishery threshold on 1 May) and for the same 12 criteria as for calculations using DS. In addition, for each water user (criterion), its annual and interval reliability, interval resilience and vulnerability were calculated.

Figures 9 and 10 show the results of WRC (water levels of Lake Baikal and releases in the Irkutsk reservoir downstream) when calculating using optimization methods for four hydrological series (two observed and two forecast).

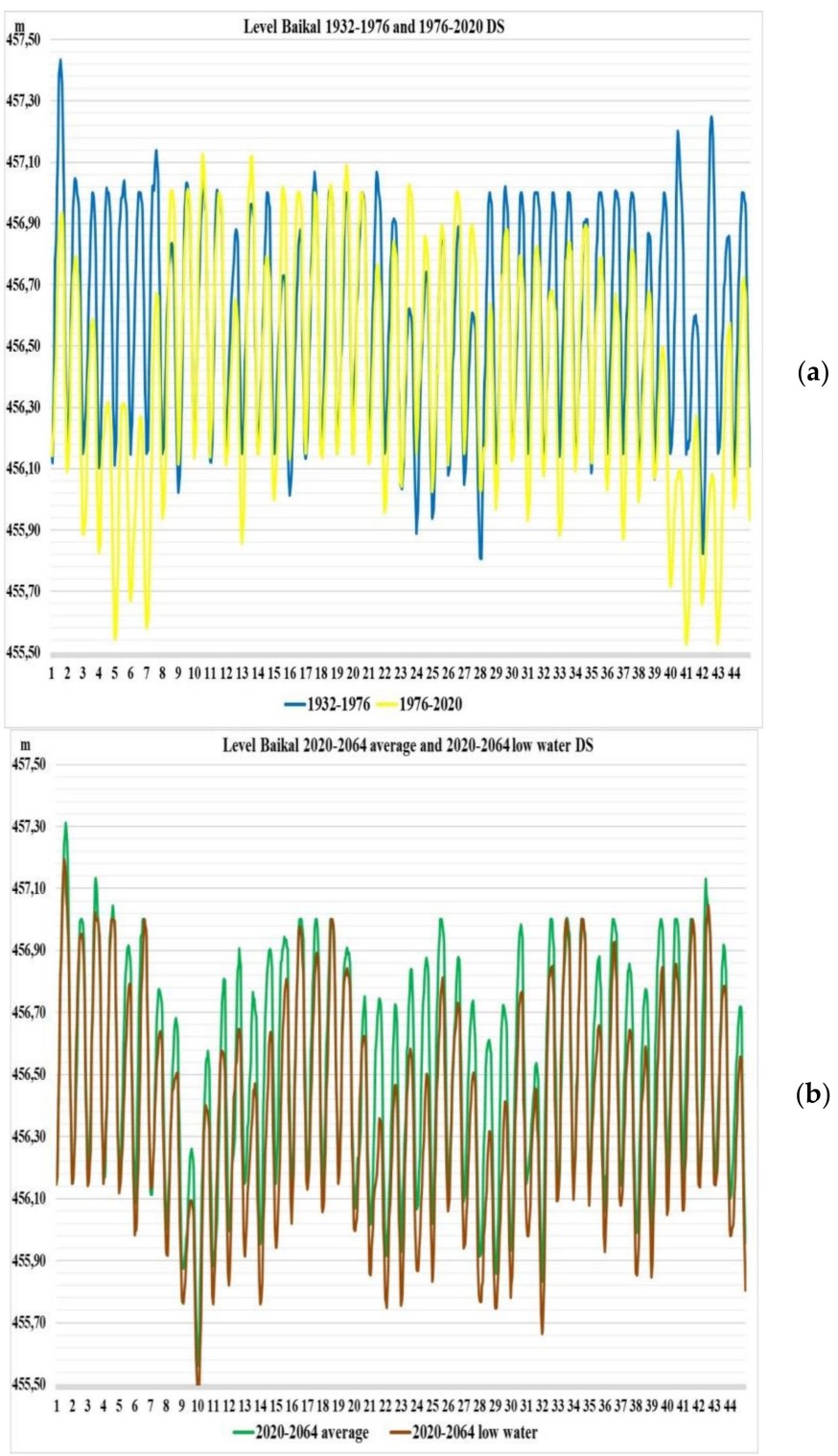

**Figure 8.** Water level of the Baikal obtained as a result of WRC by DS for series 1932–1976 and 1976–2020 (**a**) for series 2020–2064 average and 2020-2064 low water (**b**).

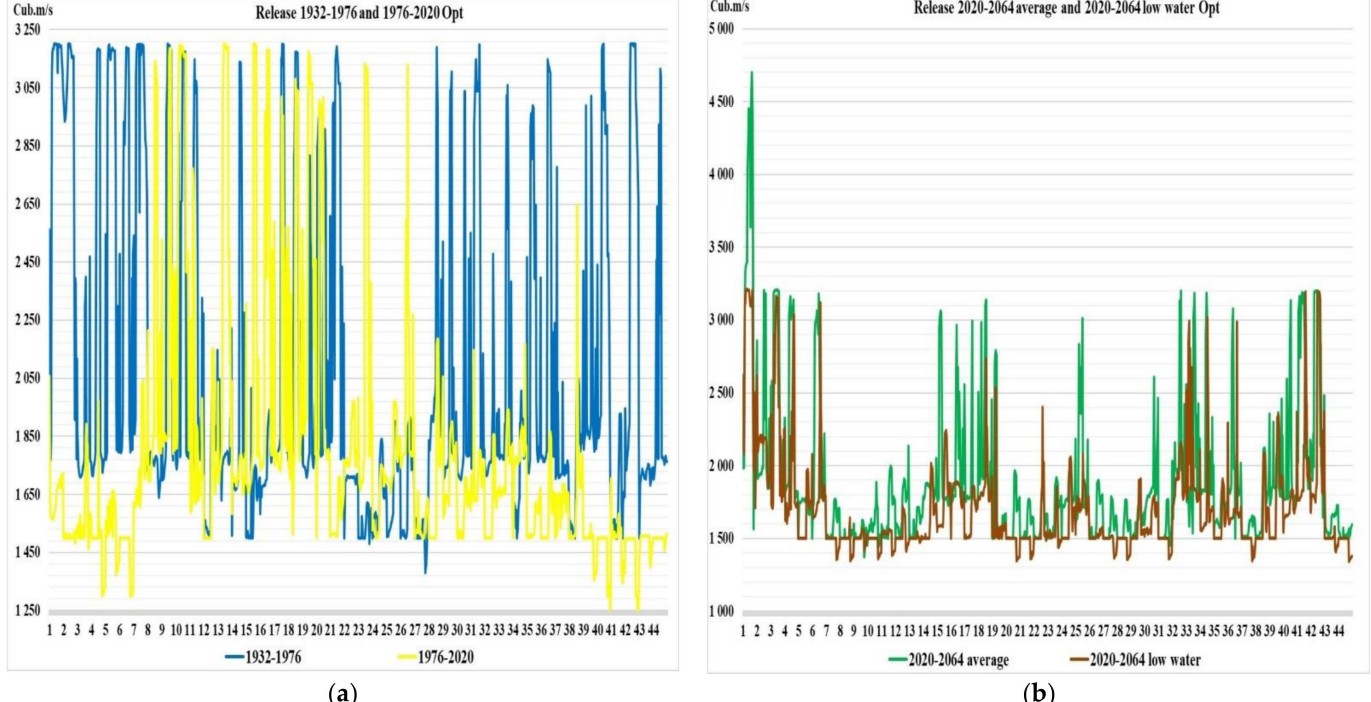

**Figure 9.** Releases to the Irkutsk reservoir downstream obtained as a result of WRC using optimization methods for series 1932–1976 and 1976–2020 (**a**) for series 2020–2064 average and 2020–2064 low water (**b**).

### 3.4. The Multi-Criteria Analysis of the Water Resource Calculations Results

The multi-criteria analysis of the water resource calculation results was performed based on the methods described in Section 2.4.

#### 3.4.1. Analysis of the Water Resource Calculations Results Using DS

Figure 11 shows column charts of annual and interval reliability for 12 criteria when performing the water resource calculations (WRC) using DS.

As can be seen from the diagrams in Figure 11a, the dispatch schedule (DS) 1988 gives good results for the 1932–1976 series on which it was built, as well as for the 2020–2064 average series, which is a forecast copy of the 1932–1976 series. Criteria 1, 5, 6, 8, 10 and 11 (Section 2.3.1) were without violations. Considering that in 1988, criterion seven had a threshold of 6000 m$^3$/s, it can be assumed that the DS 1988 made it possible to form releases with standard reliability for almost all requirements.

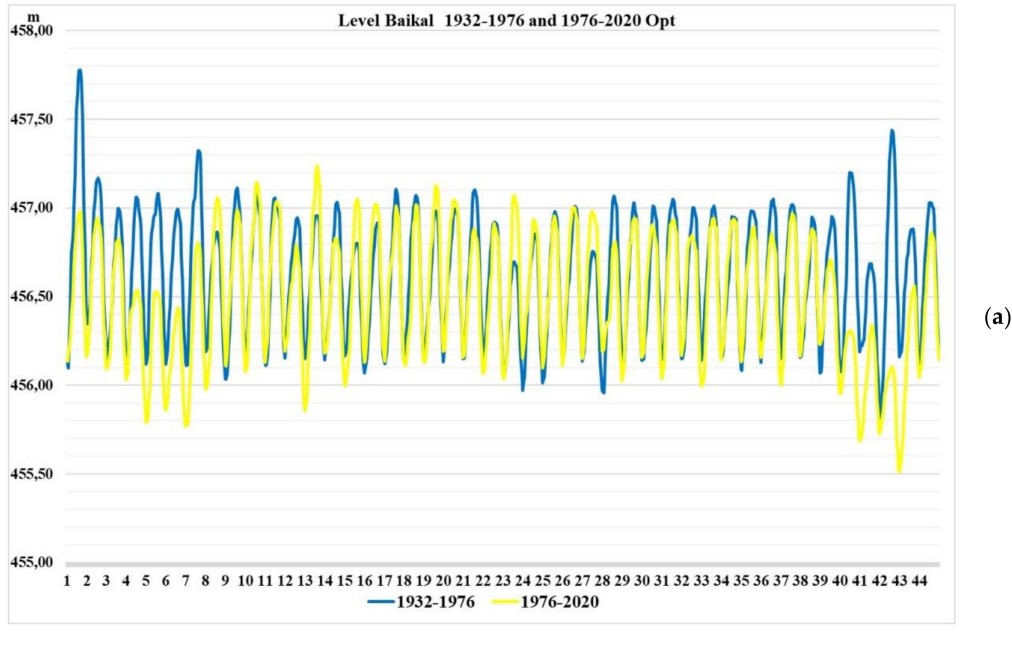

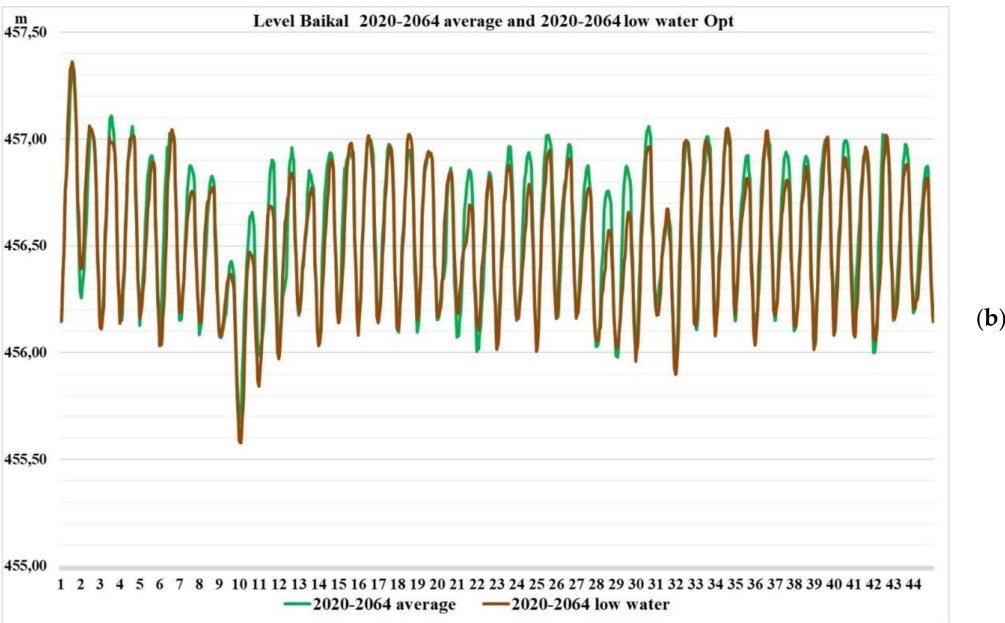

**Figure 10.** Baikal water level obtained as a result of WRC using optimization methods for series 1932–1976 and 1976–2020 (**a**) for series 2020–2064 average and 2020–2064 low water (**b**).

However, the DS 1988 cannot be used for reliable operation in modern conditions (1976–2020 series) and in forecast dry years (2020–2064 low water). The diagrams in Figure 11a shows that the DS 1988 does not provide reliable operation for criteria 1, 5, 6, 7 (3200 m$^3$/s), 8, 11 and 12. Criteria two and three were added in 2001 [25], so the DS 1988 also does not provide the required water levels.

The fishery requirements are satisfied especially "poorly", which could have caused a significant decline in the omul and whitefish population in the last 20 years. Even the interval reliability (Figure 11b), which is always greater than or equal to the annual one, does not provide the normative indicators for energy and fisheries (criteria 8, 11, 12).

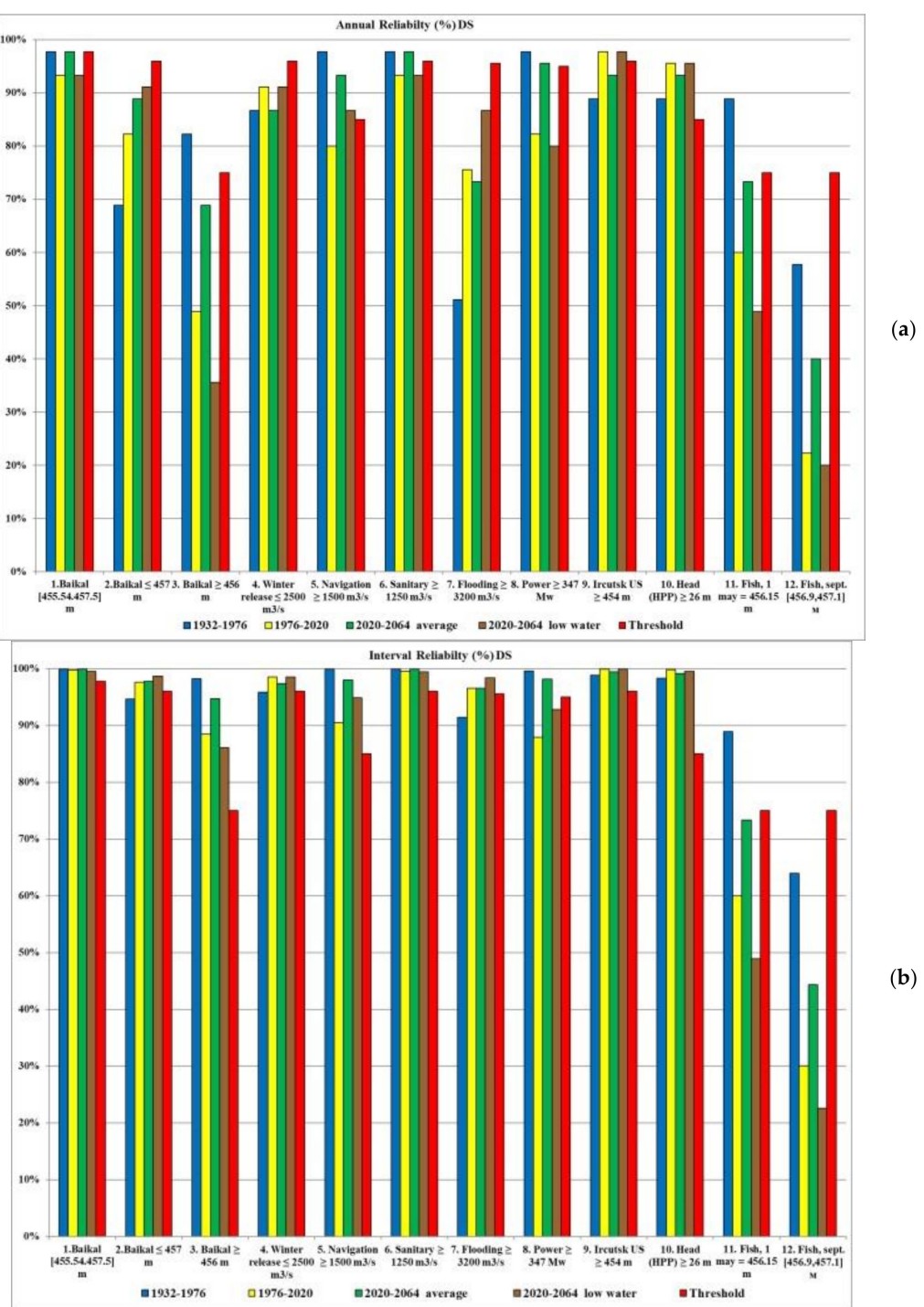

**Figure 11.** Annual (**a**) and interval (**b**) indicators of reliability when WRC uses DS for different time series.

Resilience and vulnerability diagrams (Figures 12 and 13) show that DS works without disturbances on criteria 1, 5, 6, 8 for the 1932–1976 series and has the best performance for fish criteria 11 and 12 for all the series. For the rest of the criteria and time series, the indicators are equally bad. It is also necessary to note the "bad" indicators of vulnerability.

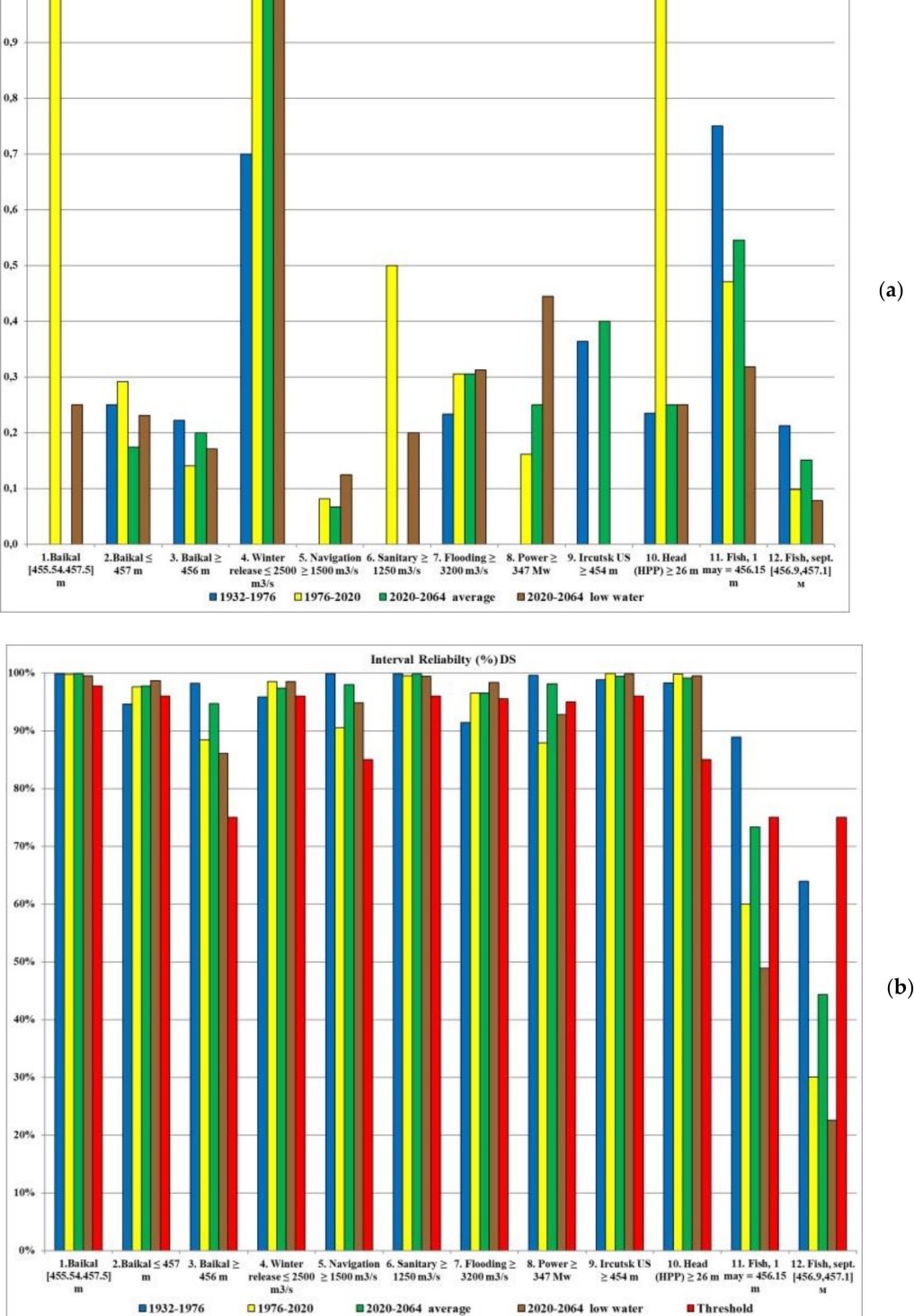

**Figure 12.** The resilience (**a**) normalized vulnerability (**b**) for different time series (DS).

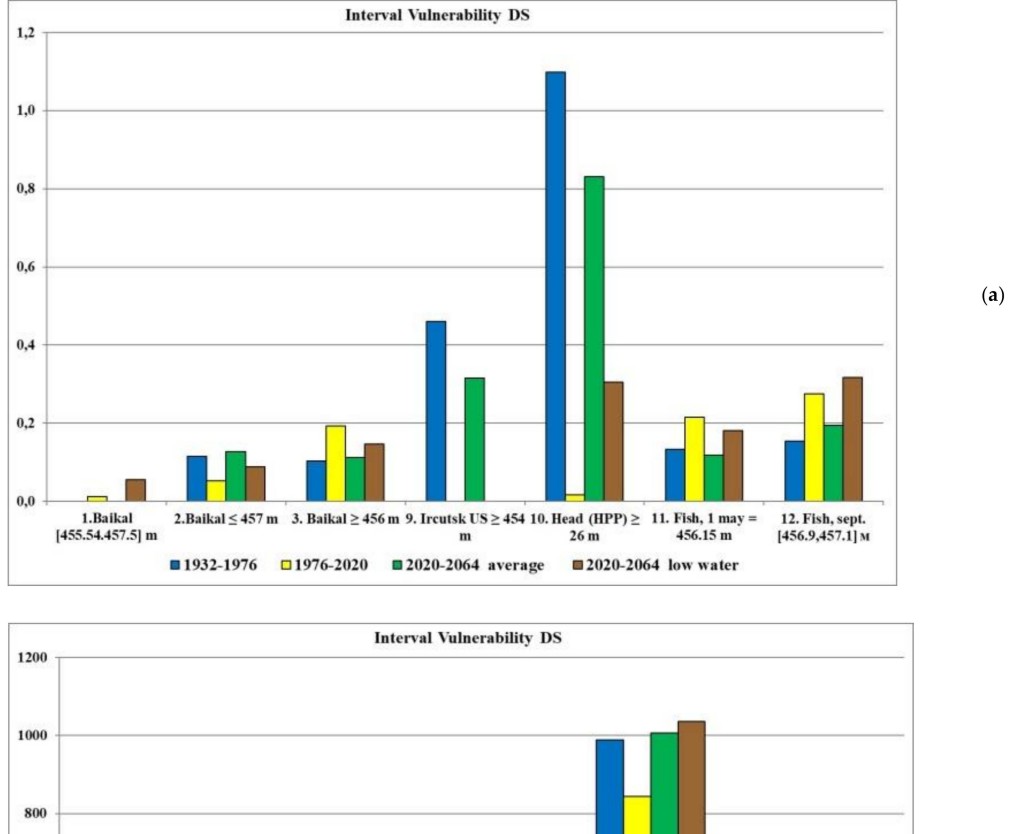

**Figure 13.** Vulnerability for different time series (DS) for criteria 1, 2, 3, 9, 10, 11, 12 (**a**) for criteria 4, 5, 6, 7, 8, (**b**).

*Conclusion:* It is necessary to rebuild DSs at least once every 10 years, taking into account the changed hydrological situation, current requirements and the water users' priorities hierarchy.

### 3.4.2. Analysis of the Water Resource Calculation Results Using Optimization Methods

The WRC results performed by optimization methods show the "water capacity" of the main river basin feeding the reservoir, and depend only on the hydrological situation, requirements and priority hierarchy of water users. Therefore, based on the WRC optimization, it is possible to analyze the genesis of retrospective and predicted hydrological inflow series. For WRCs performed on the basis of optimization methods, the approach described in Section 2.3 was used, taking into account modern requirements and the water users' priorities hierarchy (Section 2.3.1), which, unfortunately, gives a local minimum objective function (OF)

Figure 14 shows column charts (diagrams) of annual and interval reliability for 12 criteria, when performing the water resource calculations (WRC) using optimization methods.

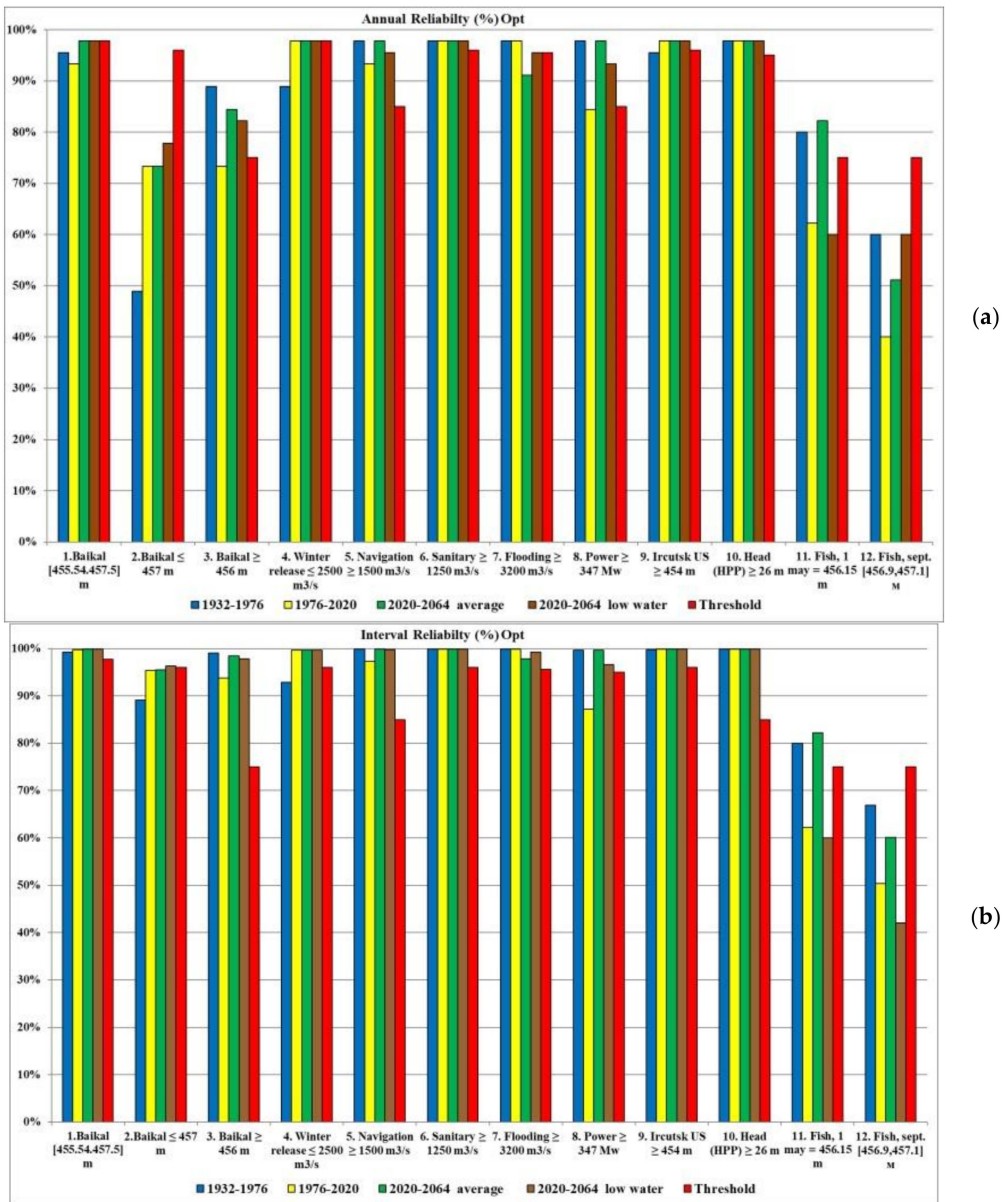

**Figure 14.** Annual (**a**) and interval (**b**) indicators of reliability when WRC use optimization for different time series.

As can be seen from the diagrams in Figure 14a, optimization significantly improves the current reliability indicators for practically all criteria and for all hydrological inflow series. Optimization allows criteria 5, 6, 9 and 10 to fulfill all time series. For the series 1932–1976, optimization allows to fulfill criteria 3, 7, 8 and 11, and criterion 1 with a slight deviation. Thus, almost all criteria are fulfilled, except 12, which cannot be fulfilled for any time series, although it has a high priority. Criterion 2 cannot be fulfilled with normative reliability for any time series, since it has been proven that the fluctuation range of Lake Baikal does not allow for keeping the water level in high-water years below 457 m. In addition, for the 1976–2020 series, it is not possible to fulfill criteria 1, 3, 8 and 11, for 2020–2064 average 7, and for the series 2020–2064 low-water 8.

It is necessary to note two facts: practically it is possible to satisfy the flood criterion 7 (3200 m$^3$/s), which is violated when using DS, and the fishery requirements are "poorly" satisfied, which will further lead to a decrease in the omul population in low-water dry years.

Interval reliability diagrams (Figure 14b) show that almost all criteria fulfil the normative indicators. However, even interval reliability is not fulfilled for the fishery (criteria 11, 12) in dry periods (1976–2020, 2020–2064 low water).

Resilience and vulnerability diagrams (Figures 15 and 16) show that optimization works without violations for almost all series on criteria 4, 6, 9, 10, with violations admissible in terms of reliability for criteria 1, 5, 7, 8. For decree no. 234 [25–27] and fisheries (criteria 2, 3, 11, 12), both in terms of the number of successive failure intervals (resilience) and the average failure value (vulnerability), the optimization does not give satisfactory results.

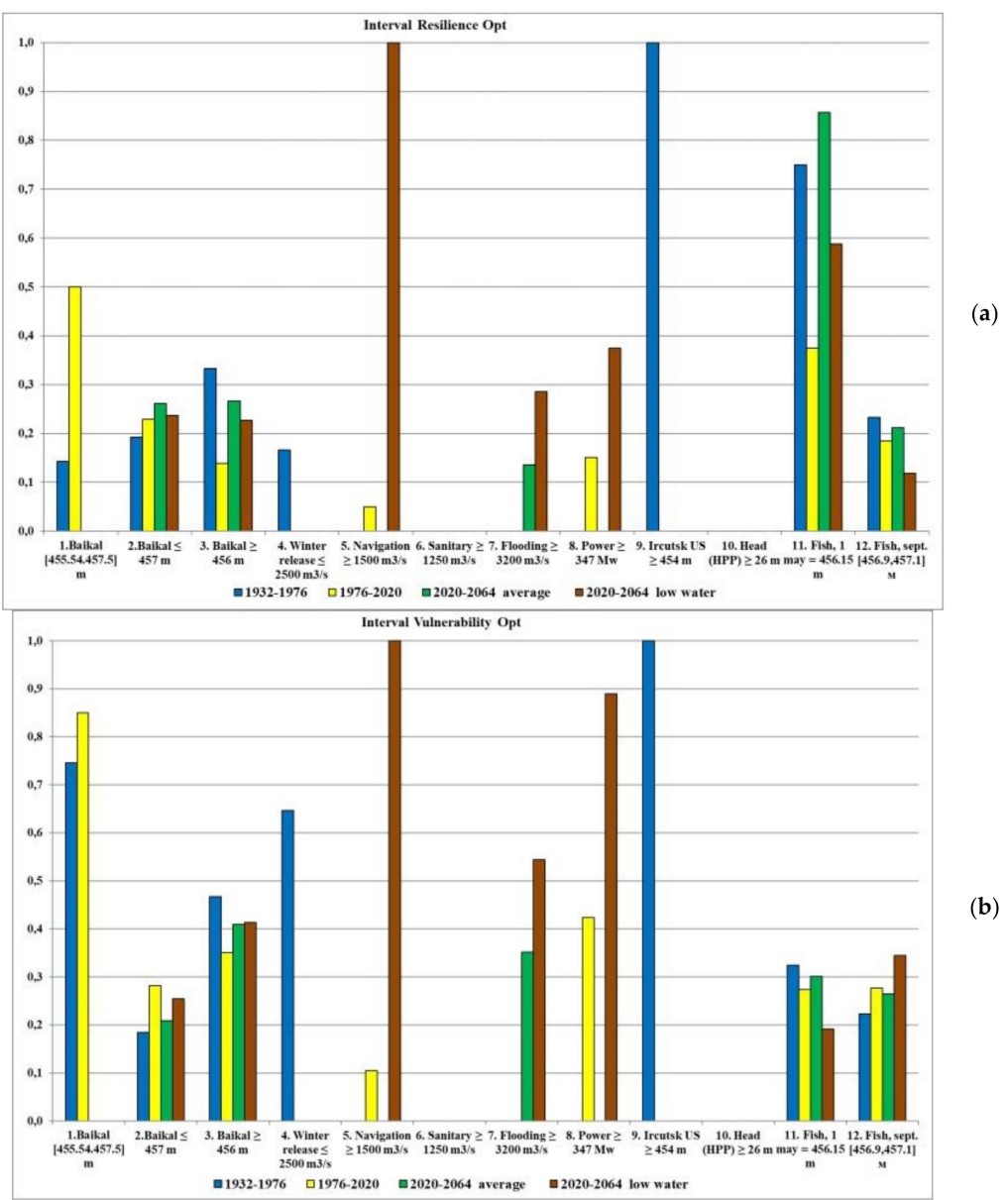

**Figure 15.** The resilience (**a**) normalized vulnerability (**b**) for different time series (optimization methods).

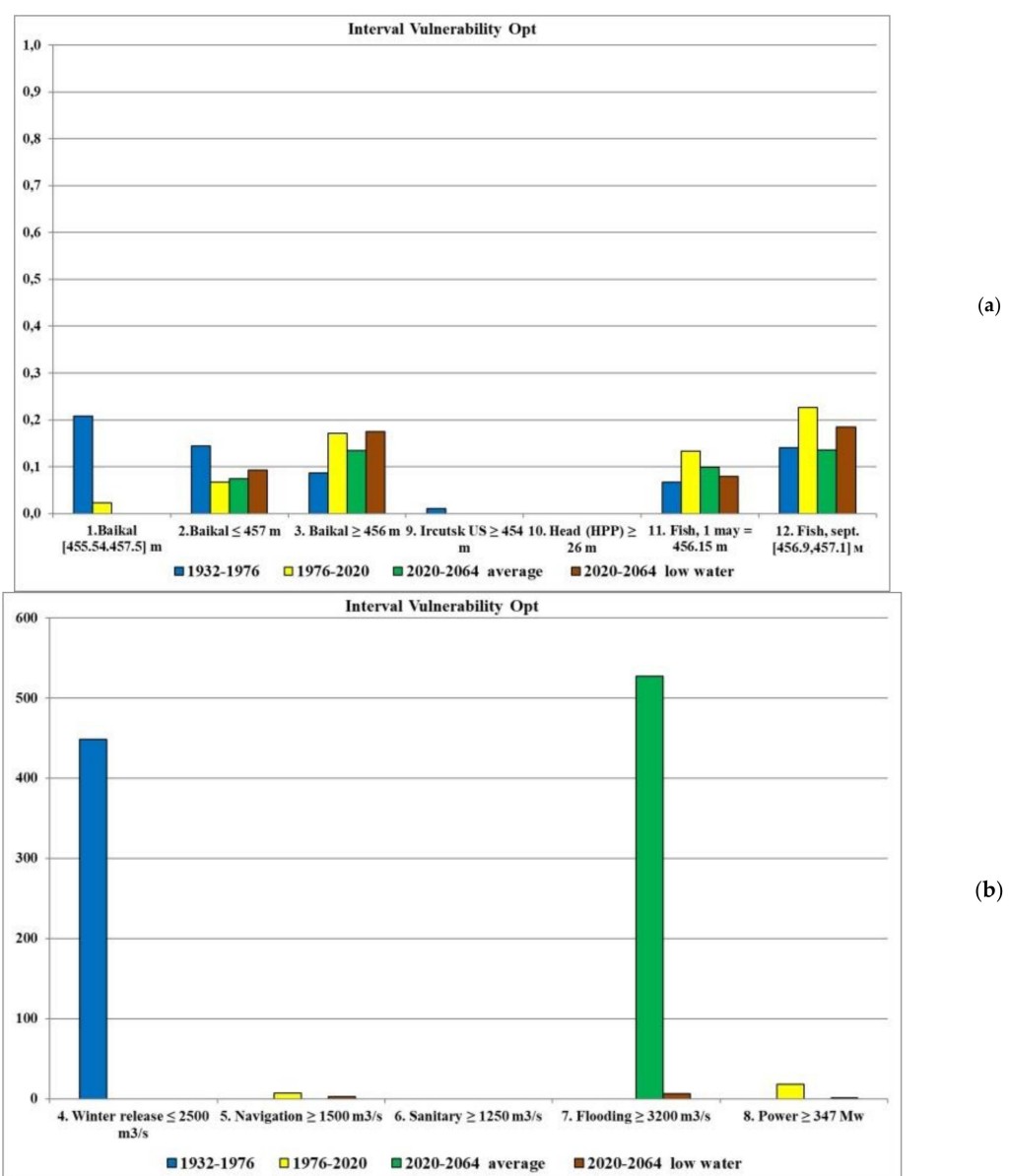

**Figure 16.** The vulnerability for different time series (optimization methods) for criteria 1, 2, 3, 9, 10, 11, 12 (**a**) for criteria 4, 5, 6, 7, 8, (**b**).

*Conclusion:* The water capacity of the Lake Baikal basin (catchment area) is sufficient to satisfy practically all water users' requirements, except for fishery, with normative reliability, even in dry periods. It is only necessary to formulate such release rules that implement the optimization approach. The use of an optimization approach for the reservoir operation mode formation significantly increases the reliability, resilience and vulnerability of management. The constructed Irkutsk hydroelectric power station violates the fishery requirements, especially in dry years, and even optimization does not allow them to be provided with sufficient reliability.

### 3.4.3. Comparison of WRC Results Obtained Using DS and Optimization Methods

Statistical performance criteria reliability, resilience and vulnerability make it possible to quantify various options for managing water resources of a reservoir under changing hydrological conditions for each individual requirement of water users. Figures 17–21 show the statistical criteria values for each water users' requirement in different hydrological conditions, 1932–1976 and 1976–2020, when performing water resource calculations for DS

1988 and using optimization methods (for forecast series 2020–2064 average and 2020–2064 low-water results are identical).

Figure 17a (series 1932–1976) shows that reliability under optimization increases for criteria 3, 4, 7, 9, 10, decreases for criterion 2 and practically does not deteriorate for all other criteria. Since the DS 1988 is tuned to the 1932–1976 series, there is no significant improvement in reliability. However, for the low-water series 1976–2020 (Figure 17b), in almost all criteria, there is a significant improvement in the reliability indicator due to the poor performance of the DS for this period.

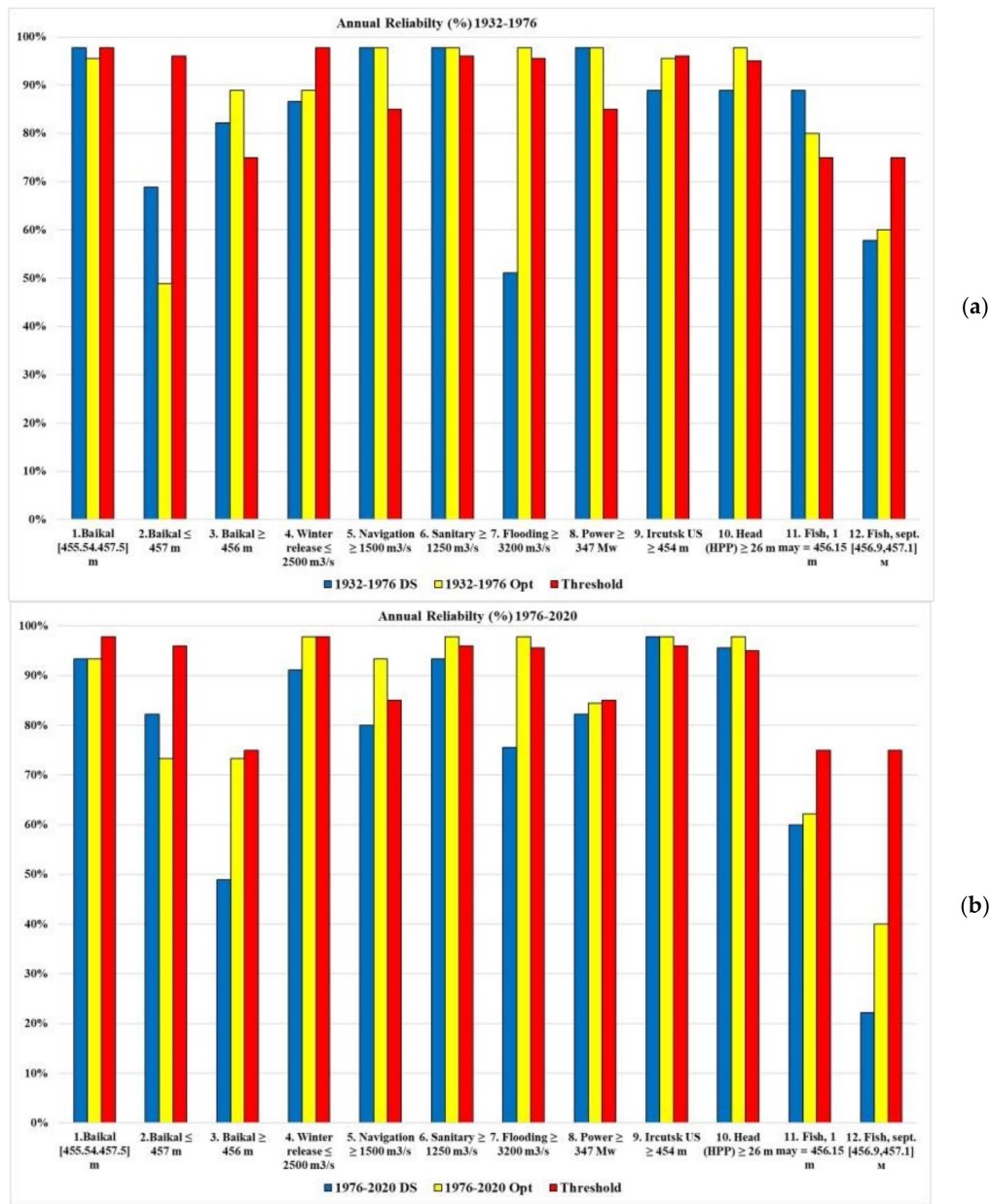

(a)

(b)

**Figure 17.** Annual reliability indicators when WRC use DS and optimization (**a**) series 1932–1976 (**b**) 1976–2020.

The use of the squared deviation from the threshold in the objective function during optimization, as a rule, leads to an increase in violations (failure), but decreases their amplitude (the maximum deviation value decreases). In this case, as a rule, there are many successive intervals with small deviations. In the years of average water content (1932–1976) it is possible to reduce by optimization such small deviations to zero, that is, to reduce the number of criteria with violations (Figure 18a). However, in dry years (1976–2020), the number of consecutive intervals with failure (resilience) increases. This is clearly seen in Figure 18b.

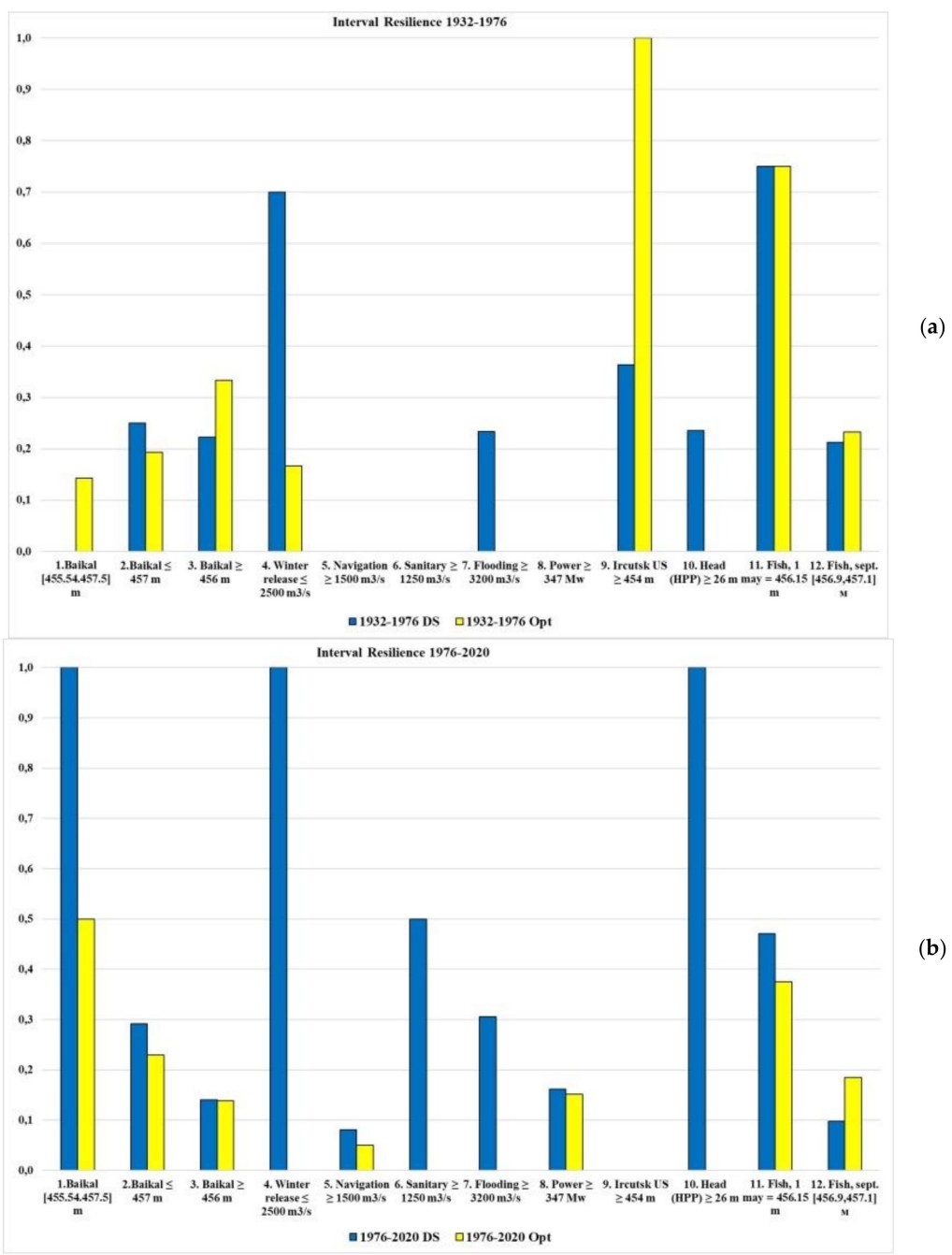

**Figure 18.** Interval resilience indicators when WRC use DS and optimization (**a**) series 1932–1976 (**b**) 1976–2020.

The above reasoning is well illustrated by Figures 19 and 20. Where it is not possible to exclude failure in optimization, the average deviation (vulnerability) value is significantly less than when using the DS for both series 1932–1976 (Figure 19) and 1976–2020 (Figure 20).

The same results are shown by the normalized interval vulnerability (Figure 21), especially for the low-water period 1976–2020 (Figure 21b).

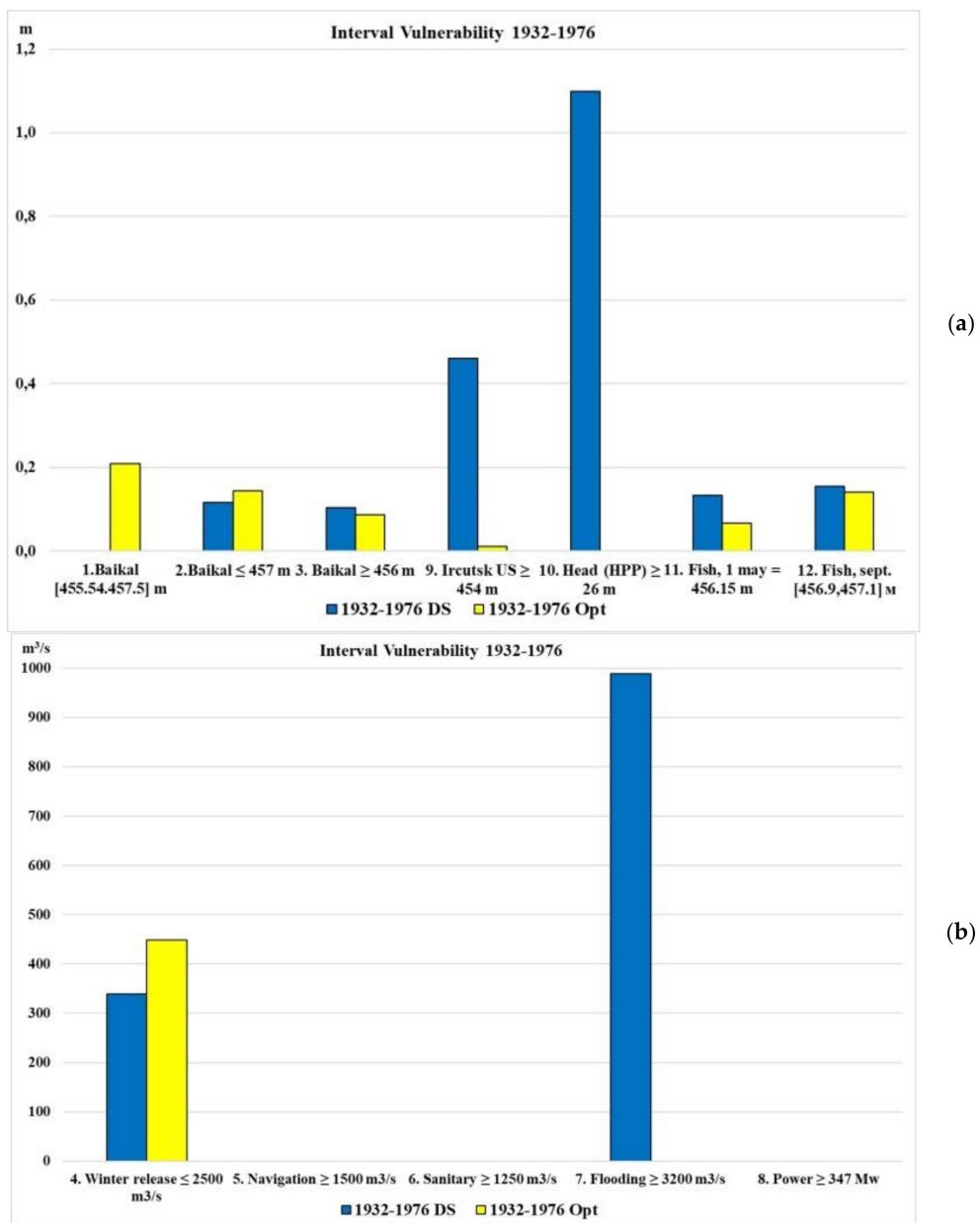

**Figure 19.** Interval vulnerability when WRC use DS and optimization (series 1932–1976) for criteria 1, 2, 3, 9, 10, 11, 12 (**a**) Figures 4–8, (**b**).

### 3.4.4. Formation of Reservoir Operation Modes Based on Optimization Methods

The carried out multi-criteria analysis (Sections 3.4.1–3.4.3) of the water resource calculation (WRC) results based on dispatch schedule (DS) and optimization methods showed that the statistical reliability characteristics obtained by optimization methods are significantly better by almost all criteria than the same indicators obtained on the basis of DS use. A legitimate question arises as to whether it is possible to create such a computational technology that would make it possible to manage the reservoir water

resources in an operational mode on the basis of optimization methods and inflow forecast for this period (short-term and long-term forecast)?

Due to possible climate change, the use of DSs, which, as a rule, are updated every ten years (or even less frequently) and are based on long-term retrospective series of inflows over a long period, can lead to serious errors in management. This is confirmed by the fact that during high floods, as a rule, the reservoir is managed on the basis of operator's professional experience, without using DSs.

The second important factor that affects the reliability of DS management is the change over time in the water user requirement priorities (see Section 2.3.1). This too cannot be taken into account when constructing the DS. Calculations have shown that the management using the DS 1988 in the "Lake Baikal–Irkutsk Reservoir" has worse reliability indicators for the 1976–2020 inflow series than for the 1932–1976 series on the basis of the series and the requirements during the period in which it was built, not only because the climate has changed, but also because of the changed priorities.

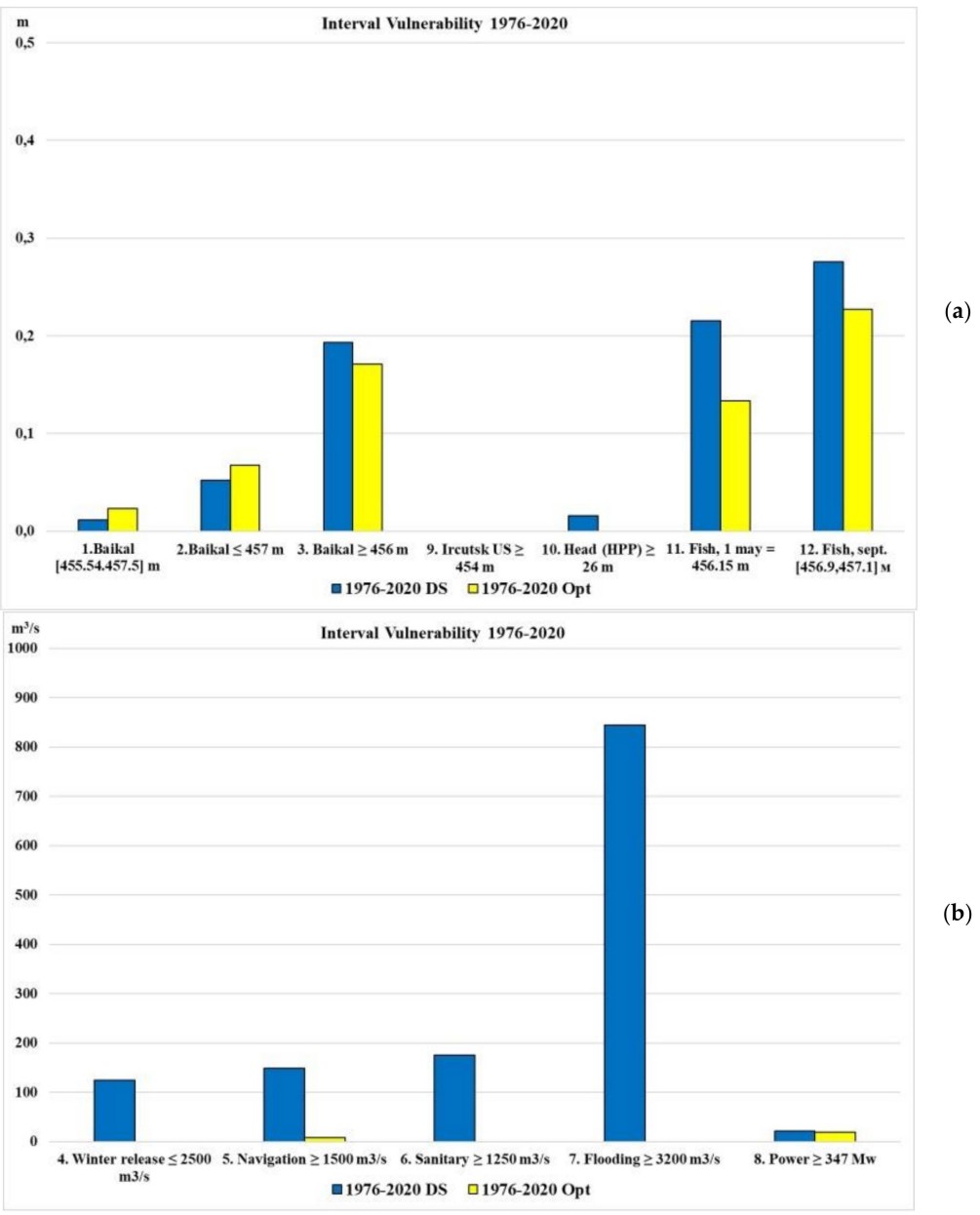

**Figure 20.** Interval vulnerability when WRC use DS and optimization (series 1976–2020) for criteria 1, 2, 3, 9, 10, 11, 12 (**a**) for criteria 4, 5, 6, 7, 8, (**b**).

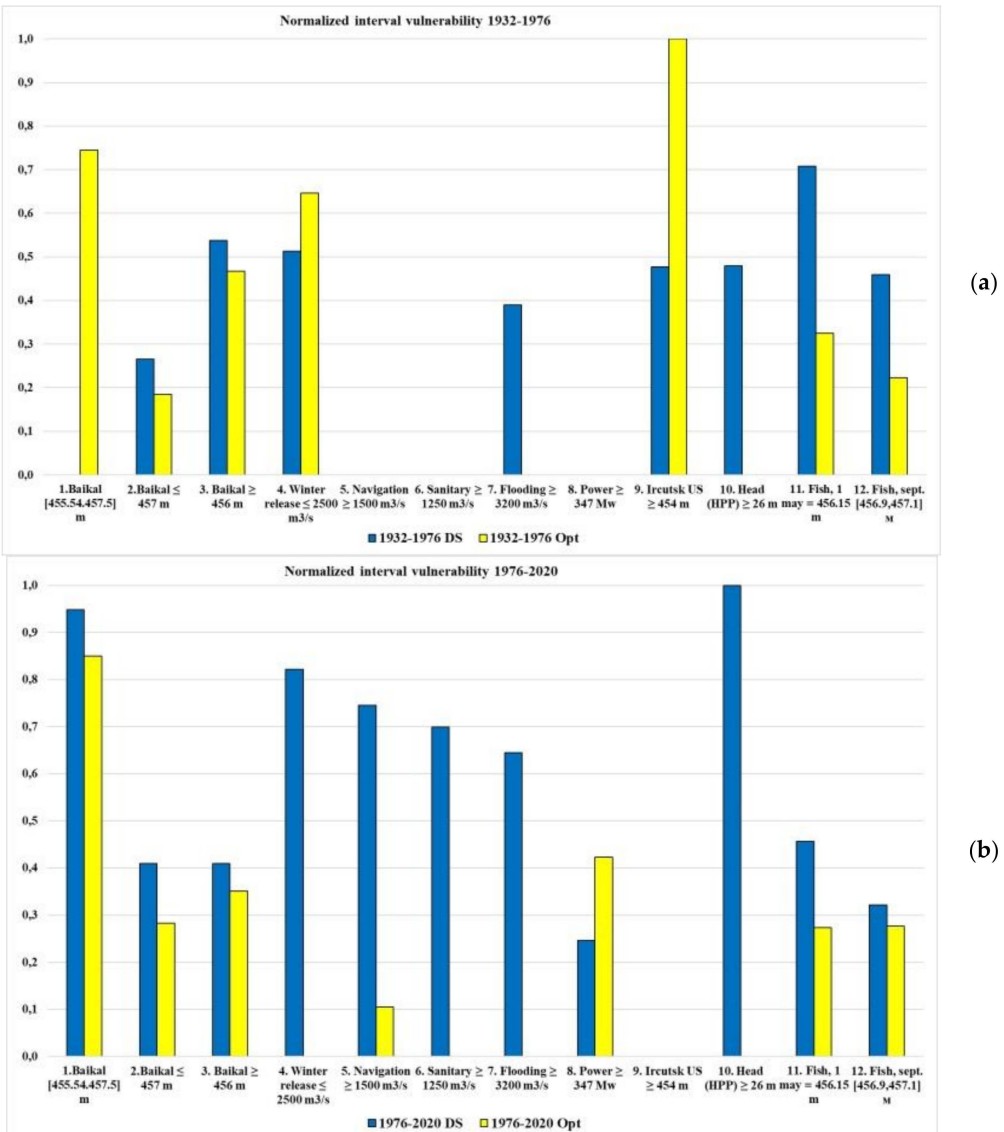

**Figure 21.** Interval vulnerability when WRC use DS and optimization (**a**) series 1932–1976 (**b**) series 1976–2020.

There are numerous publications on the optimization when performing water resource calculations; however, the authors were unable, unfortunately, to find publications containing optimization approaches to manage reservoir water resources in an on-line mode, based on optimization methods and the trade-off theory. The authors proposed this approach. Below is a mathematical model and an algorithm of such a computational technology developed by the authors.

The strict meaningful formulation of the problem is as follows: to create a computational technique that would make it possible to manage the reservoir water resources on-line, based on optimization methods without using DS. That is, form releases to the reservoir downstream for the next time interval (usually 10–30 days) depending on the reservoir volume (level) at the interval beginning; the short-term and long-term hydrological inflow forecast for this interval; the long-term observed inflow series for the previous years; and the actual water users' requirement priorities hierarchy for this period. The time interval duration depends on the hydrological situation during this period: summer-autumn or winter low-water period, spring or rain floods.

Let there be a realized hydrological series $\mathbf{P}$ of inflow to a reservoir with a duration of $\mathbf{T}$ years. It is necessary to make a decision on the reservoir operating modes in the current $(\mathbf{T} + \mathbf{1})$th year.

Each year is divided into $\mathbf{m}$ estimated time intervals (pentads, decades, months) in accordance with the formed hydrological observed inflow. For the Lake Baikal, every month from May to October is divided into three conditional decades (the last one can contain 11 days), and the rest of the intervals are monthly (24 intervals in total). For the last interval of $\mathbf{m}$ year of $\mathbf{T}$, the reservoir volume $\mathbf{W^f_{TM}}$, observed at the end of the interval is known. This volume is taken as equal to the initial volume of $\mathbf{W^b_{T+11}}$ reservoirs in the first estimated time interval of the $(\mathbf{T} + \mathbf{1})$th year. Below we describe an algorithm finding an optimal solution.

1.  We construct a predicted long-term hydrological series of inflows $\mathbf{P^f}$ for the last 40–50 years of observed inflows with a duration of $\mathbf{T^f}$ years, using the method described in Sections 2.1 and 3.1 or other methods [16].

$$P^f = \{ P^f_{tm} \mid t = [1,T^f], m = [1,M]\} \tag{25}$$

2.  **The algorithm is built on the principle of mathematical induction.** The basis of induction is as follows: the calculation for the first interval $\mathbf{m} = \mathbf{1}$, year $\mathbf{t} = \mathbf{1}$ (corresponds to $\mathbf{T} + \mathbf{1}$ year of the original series $\mathbf{P}$). The reservoir volume at the beginning of the $\mathbf{W^b_{11}}$ interval is known and is equal to $\mathbf{W^f_{TM}}$. For the first interval, we set a hierarchy of priorities for water users' requirements, which are actual for this calculation. Based on the mid-term interval inflow forecast, we determine the value $\mathbf{P^{f^*}_{11}}$ and replace $\mathbf{P^f_{11}}$ на $\mathbf{P^{f^*}_{11}}$, $\mathbf{P^f_{11}} := \mathbf{P^{f^*}_{11}}$ (if the forecast is deeper, then replace the inflow in $\mathbf{P^f}$ for other intervals $\mathbf{m} > \mathbf{1}$). After that we solve the optimization task using the hydrological series $\mathbf{P^f}$ with the replaced predicted inflow for given priorities, with a given $\mathbf{W^b_{11}}$ by the algorithm given in Section 2.3.1. Variables that are optimized during the calculation are the releases $\mathbf{R_{tm}}$, where $\mathbf{t} = [\mathbf{1,T^f}]$, $\mathbf{m} = [\mathbf{1,M}]$. The found solution, $\mathbf{R_{11}}$ is taken as the desired (or as the sought for) one $\mathbf{R_{T+11}}$. After the solution implementation in the management process, the actual reservoir volume at the end of the first interval $\mathbf{W^f_{T+11}}$, is determined (real $\mathbf{R^*_{T+11}}$ may differ from $\mathbf{R_{11}}$) which is taken as the initial volume in the second interval $\mathbf{W^b_{T+12}} = \mathbf{W^f_{T+11}}$. The predicted inflow $\mathbf{P^{f^*}_{11}}$ is replaced by the actual observed $\mathbf{P^{f^{**}}_{11}}$ and is stored instead of $\mathbf{P^f_{11}}$.

3.  **The induction step.** Assume that the reservoir operating modes for the interval $\mathbf{m} > \mathbf{1}$ are formed and the initial volume $\mathbf{W^b_{T+1m+1}}$ is determined. The search algorithm of releases $\mathbf{R_{T+1m+1}}$ for the interval $\mathbf{m} + \mathbf{1}$ is similar to step two, i.e., for the interval $\mathbf{m} + \mathbf{1}$, we set a hierarchy of priorities for water users' requirements, which are actual for this calculation. Now, based on the short-term interval inflow forecast, we determine the value $\mathbf{P^{f^*}_{1m+1}}$ and replace $\mathbf{P^f_{1m+1}}$ на $\mathbf{P^{f^*}_{1m+1}}$, $\mathbf{P^f_{1m+1}} := \mathbf{P^{f^*}_{1m+1}}$ (if the forecast is deeper, then replace the inflow in $\mathbf{P^f}$ for other intervals $>\mathbf{m} + \mathbf{1}$). Then we solve the optimization task using the hydrological series $\mathbf{P^f}$ with the replaced predicted inflow for given priorities with a given $\mathbf{W^b_{1m+1}}$ by the algorithm given in Section 2.3.1. Variables that are optimized during the calculation are releases $\mathbf{R_{tm}}$, where $\mathbf{t} = [\mathbf{1,T^f}]$, $\mathbf{m} = [\mathbf{1,M}]$. The found solution $\mathbf{R_{1m+1}}$ is taken as the desired one $\mathbf{R_{T+1m+1}}$. After solution implementation in the management process, the actual reservoir volume at the end of the first interval $\mathbf{W^f_{T+1m+1}}$, is determined (real $\mathbf{R^*_{T+1m+1}}$ may differ from $\mathbf{R_{1m+1}}$), which is taken as the initial volume in the second interval $\mathbf{W^b_{T+1m+2}} = \mathbf{W^f_{T+1m1}}$. The predicted inflow $\mathbf{P^{f^*}_{1m+1}}$ is replaced by the actual observed $\mathbf{P^{f^{**}}_{1m+1}}$ and is stored instead of $\mathbf{P^f_{1m+1}}$. Step three is repeated until $\mathbf{m} = \mathbf{M}$.

4.  If necessary, the daily value release in the interval m is determined, so that the average release for the interval is $\mathbf{R_{T+1m}}$.

5.  The initial series $\mathbf{P}$ is supplemented by the observed inflow data for the year $\mathbf{T} + \mathbf{1}$. The steps 1–5 are repeated for the new series $\mathbf{P}$ and the next year $\mathbf{T} + \mathbf{2}$, etc.

## 4. Discussion

Statistical performance criteria (reliability, resilience, vulnerability) when carrying out water resource calculations make it possible to quantify the reliability of using the adopted release rule in relation to a given hydrological series of inflows for each individual requirement of water users. How does one assess the complex impact of all the criteria, defining the requirements of water users on the release rule quality?

Unfortunately, the additive objective function (OF) gives a poor representation of the quality and total impact of the release rule on the comprehensive (by all criteria) fulfillment of the requirements of water users. The sum of failures for all criteria does not give such representation. Both variants of the assessment show only a relative improvement or deterioration of the release rule in relation to different time series of inflow. For example, the total number of failures in the given eight calculations shows (Table 1) that optimization significantly reduces the number of failures (by 1.5–2 times), but, it is impossible to determine whether this improves the reliability of the high priority criteria.

**Table 1.** Comparison by the sum of failures of the WRC using DS and optimization on different time series of inflow.

| Time Series of Inflow | WRC Using DS | WRC Using Optimization | Improvement% |
|---|---|---|---|
| Annual Reliability (Failure) | | | |
| 1932–1976 | 92 | 57 | 38% |
| 1976–2020 | 121 | 74 | 39% |
| 2020–2064 average | 83 | 48 | 42% |
| 2020–2064 low water | 117 | 63 | 46% |
| Interval reliability (failure) | | | |
| 1932–1976 | 321 | 200 | 38% |
| 1976–2020 | 431 | 249 | 42% |
| 2020–2064 average | 261 | 142 | 46% |
| 2020–2064 low water | 387 | 169 | 56% |

For the release rule multi-criteria analysis for various time series and water users' requirements, two integral statistical criteria are proposed: the integrated normalized reliability index (**INRI**) and the integrated normalized vulnerability index (**INVI**).

*The integrated normalized reliability index* (**INRI**) is defined as the sum of failure for all criteria, divided by the number of years in the time series. It is determined by the formula**:**

$$\text{INRI} = \Sigma_{k\,=\,[1,K]}\,(1\text{—Reliability}^{k}[\mathbf{X}]) \tag{26}$$

where reliabilty$^{k}$[**X**] is the reliabilty (in fractions) for the $k$th criterion, K is the number of criteria.

The **INRI** can be determined for both annual and interval reliability, but it is better to use the annual **INRI**, since there are normative values for the reliability of the criteria for it. This normative reliability can be used as an estimated **T$_{INRI}$** threshold for **INRI** by defining it as follows:

$$\mathbf{T_{INRI}} = \Sigma_{k\,=\,[1,K]}\,(1-\mathbf{T}^{k}) \tag{27}$$

where **T**$^{k}$ is the normative security for the $k$th criterion expressed in fractions.

**INRI** characterizes the control components (a set of criteria, release rule and inflow time series) as follows: $\mathbf{0 \le INRI \le K}$; the closer the **INRI** value is to **T$_{INRI}$**, the better the control components are combined, if the **INRI $\le$ T$_{INRI}$** value, then the release rule can be used for management. Although it is impossible to set the absolute **INRI** indicator value, which characterizes the release rule quality for a given set of water users' requirements and inflow time series, this indicator allows you to form different management options for different criteria and inflow time series, to compare them with the **T$_{INRI}$**, in order to evaluate and thus to choose the best solution. Table 2 shows the annual **INRI** values for eighth performed WRC.

**Table 2.** INRI indicator values for the eight performed WRCs.

| Time Series of Inflow | WRC Using DS | WRC Using Optimization | Improvement% |
|---|---|---|---|
| 1932–1976 | 1.96 | 1.53 | 22% |
| 1976–2020 | 2.78 | 1.91 | 31% |
| 2020–2064 average | 1.98 | 1.33 | 33% |
| 2020–2064 low water | 2.80 | 1.47 | 48% |
| $T_{INRI}$ | 1.33 | 1.33 | |

It can be seen from Table 2 that DS works best for time series 1932–1976, 2020–2064 average and worse for 1976–2020, 2020–2064 low water. The results of optimization calculations are all much better than calculations for DS, and for the time series 2020–2064 the average is **INRI** = **$T_{INRI}$**.

*The integrated normalized vulnerability index* **(INVI)** is defined as the sum of the normalized vulnerabilities for all criteria and is determined by the formula:

$$\mathbf{INVI} = \Sigma_{k\,=\,[1,K]}\text{Vulnerability\_N}^k[\mathbf{X}]) \tag{28}$$

where vulnerability_$N^k[\mathbf{X}]$ is the normalized vulnerability for the kth criterion.

**INVI** characterizes the control components (a set of criteria, release rule and inflow time series) as follows: **$0 \leq INVI \leq K$**, when the average deviation is equal to the maximum, then **INVI** shows the number of criteria without violations (failure), indicator **$T_{INVI}$ = INVI/K** gives the fraction of the average value faliure from the maximum for the entire set of criteria. The indicator $T_{INVI}$ makes it possible to estimate the average depth of failure, depending on the maximum for the entire set of criteria, therefore, frpm knowing the maximum depth of failure for the criterion, one can decide on the significance of this failure and depending on the specific requirement, neglect such failure.

Although, as for **INRI**, it is impossible to set the absolute **INVI** indicator value, which characterizes the release rule quality for a given set of water users' requirements and inflow time series, however, this indicator allows you to compare different release rule options for different criteria and inflow time series, and the **$T_{INVI}$** value makes it possible to estimate the depth of failures and, at small values, to neglect them. Table 3 shows the **INVI** and **$T_{INVI}$** values for the performed eighth WRC.

**Table 3.** Values of the INRI indicator for the eight performed WRCs.

| Time Series of Inflow | WRC Using DS | WRC Using Optimization | Improvement % | $T_{INVI}$ (DS) | $T_{INVI}$ (Opt) |
|---|---|---|---|---|---|
| 1932–1976 | 3.83 | 3.37 | 12% | 0.32 | 0.28 |
| 1976–2020 | 6.70 | 2.28 | 66% | 0.56 | 0.19 |
| 2020–2064 average | 5.04 | 1.27 | 75% | 0.42 | 0.11 |
| 2020–2064 low water | 5.78 | 3.29 | 43% | 0.48 | 0.27 |

Table 3 shows that DS works best for time series 1932–1976, and worse for all other series. The results of optimization calculations are all much better than calculations for DS, and **INVI** for the time series 2020–2064 average is generally extremely low. The fraction of the cumulative average failure depth is 0.11. Figure 22 shows the calculation result comparison (normalized vulnerability) for the series 2020–2064 average for DS and optimization methods. The management quality by optimization methods is obvious.

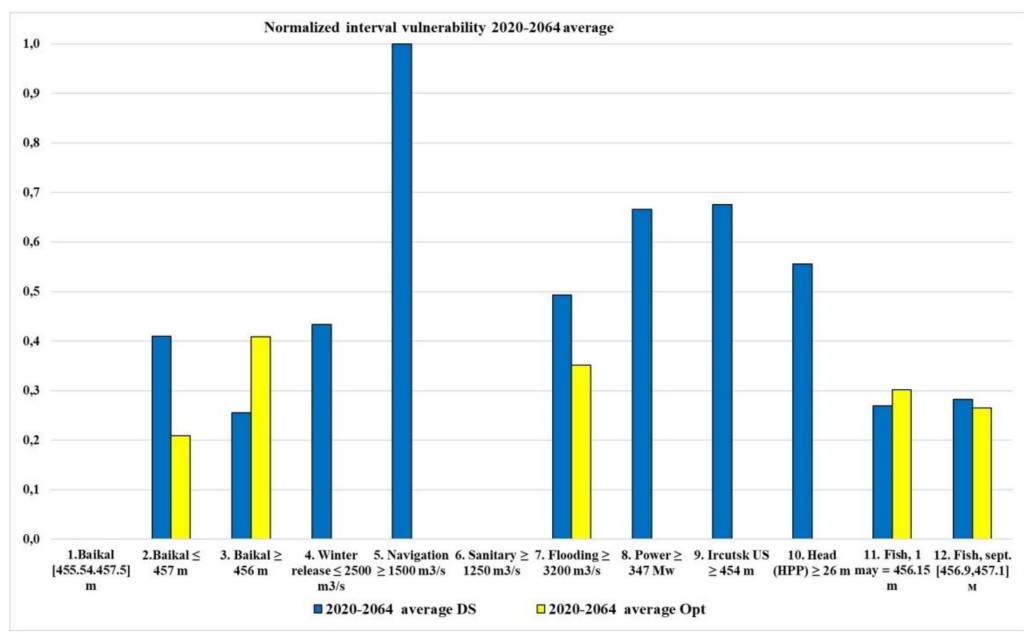

**Figure 22.** Normalized vulnerability when WRC use DS and optimization for the time series 2020–2064 average.

## 5. Conclusions

The article presents the research results on a justified operating mode formation of the "Lake Baikal–Irkutsk reservoir" complex, for various retrospective and forecast time series of inflow and various release rules, considering the real hierarchically ordered requirements of water users. Water resource calculations were carried out using the dispatch schedule, approved since 1988, and on the basis of optimization methods. An optimization method based on the decomposition of the optimization taskinto a number of subtasks of lesser dimension (two years) has been developed especially for performing water resource calculations. The calculations were carried out for four time series of inflows: observed 1932–1976; observed 1976–2020; predicted 2020–2064, built according to the average water content period statistical parameters of the observed inflow time series; forecast 2020–2064, built according to the low-water period statistical parameters of the observed inflow time series. A mathematical model is presented and predictive hydrological series of various genesis are formed. Based on the performed water resource calculations, a multi-criteria analysis of the modeling results was carried out. Based on the research carried out, the following conclusions can be drawn.

1. The dispatch schedule of 1988 does not give reliable results when performing water resource calculations on modern hydrological series, in comparison with the inflow series on the basis of which it is built. This suggests, on the one hand, that the inflow genesis has changed (there was an intra-annual change in runoff), on the other hand, over the past 44 years, the average annual inflow has decreased significantly (by 13%). The average annual inflow for the 1932–1976 series was 63.7 km$^3$, and for the 1976–2020 series-55.7 km$^3$. Calculations have shown that if a low-water forecast is implemented, reliability, resilience and vulnerability of water users will significantly deteriorate when using DS 1988. For normal operation of the Irkutsk reservoir, it is necessary to develop a new dispatch schedule that would consider modern hydrology (last 20–30 years), modern requirements of water users and modern priorities (such attempts were undertaken in 2004, 2007 and 2013, however, new DSs were not approved due to the conflict between the water users' requirements during these years).

2. As shown by multi-criteria analysis, water resource calculations based on optimization methods give results in terms of reliability, resilience and vulnerability, much better than when using DS. However, in almost all studies, optimization is used only for

strategic planning of measures to improve the water resource situation in the river basin. Therefore, the authors have developed a mathematical model, an algorithm and computer technique for the formation of the reservoir optimal trade-off operation modes in real-time (for the next time interval) based on optimization methods. The data for calculating releases are: reservoir volume at the beginning of the interval, short-term (for the next interval) and long-term hydrological forecast, and given hierarchically ordered requirements of water users. For the implementation of the computer technique, a unique optimization algorithm was developed that allows for quick solving of complex nonlinear tasks of large dimensions. This approach will significantly improve the parameters of the reliability of management decisions in comparison with DS.

3. Methods for a comprehensive assessment of the developed rules for reservoir management (release rule) are proposed, based on an integrated analysis of the set water users' requirements (criteria) when assessing the management reliability.

4. As a discussion, a long-term forecasting method based on a fairly good correlation between precipitation in a particular region, and the 11-year cycle of the Schwab solar activity is proposed.

**Author Contributions:** Conceptualization, A.B. and M.B.; methodology, A.B.; software, A.B.; validation, A.B. and M.B.; formal analysis, A.B. and M.B.; investigation, A.B. and M.B.; resources, A.B.; data curation, A.B.; writing—original draft preparation, A.B. and M.B.; writing—review and editing, A.B. and M.B.; visualization, A.B.; supervision, A.B.; project administration, M.B.; funding acquisition, M.B. A.B.—Abstract, Sections 1 and 2 except Sections 2.1 and 3 except Sections 3.1, 4 and 5. M.B.—Abstract, Sections 1, 2.1, 3.1 and 5. All authors have read and agreed to the published version of the manuscript.

**Funding:** The present work was carried out within the framework of the Governmental assignment of the Water Problems Institute of the Russian Academy of Sciences no. 0147-2019-0003 (AAAA-A18-118022090105-5). Russian Foundation for Basic Research (RFBR), Grant 17-29-05108 ofi_m.

**Institutional Review Board Statement:** Not applicable.

**Informed Consent Statement:** Not applicable.

**Data Availability Statement:** The text of the manuscript contains links to data (http://pivr.enbvu.ru, http://www.rushydro.ru/hydrology/informer/ accessed on 11 October 2021).

**Acknowledgments:** The results outlined in the paper were obtained with the financial support of the Russian Foundation for Basic Research (RFBR), Grant 17-29-05108 ofi_m.

**Conflicts of Interest:** The authors declare no conflict of interest. The funders had no role in the design of the study; in the collection, analyses, or interpretation of data; in the writing of the manuscript, or in the decision to publish the results.

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
