# Peer review of "Multi-Criteria Analysis of the “Lake Baikal—Irkutsk Reservoir” Operating Modes in a Changing Climate: Reliability, Resilience, Vulnerability"

_water, doi:10.3390/w13202879_

Round 1

Reviewer 1 Report

It is an interesting work, but in many cases is difficult to understand the manuscript. I suggest that the English language should be carefully checked in the entire document.

Since the paper deals with the "lake Baikal - Irkutsk Reservoir" operating modes in a changing climate, my major concern is that there is almost no reference about evaporation (the term evaporation is found 1 time in the manuscript). Is evaporation and the potential changes due to climate change are taken into account and how?

Please check the attached file to see some other minor comments on the highlighted text.

Author Response

Response to Reviewer 1 Comments

Point 1: It is an interesting work, but in many cases is difficult to understand the manuscript. I suggest that the English language should be carefully checked in the entire document.

Response 1: We have carefully checked the English language throughout the document. We attracted two native speakers: from the USA and from Australia - and a translator from the VNIIGiM institute. Many changes have been made.

Point 2: Since the paper deals with the "lake Baikal - Irkutsk Reservoir" operating modes in a changing climate, my major concern is that there is almost no reference about evaporation (the term evaporation is found 1 time in the manuscript). Is evaporation and the potential changes due to climate change are taken into account and how?

Response 2: We have included explanations in the text of the article.Lake Baikal is a unique lake, 334 rivers inflow into it and only one outflow - the Angara river. Therefore, the calculation of the inflow to the lake is carried out by the balance method. For the average level, bathymetric functions determine the volume at the begin and at the end of the time step. To their difference is added release from the Irkutsk reservoir - this will be an inflow. In Russia, the inflow calculated by the balance equation is called the useful inflow. Naturally, it takes into account evaporation and seepage into groundwater, so it is not necessary to add them to the balance equations. In the balance equations given in the article, a lot of things are missing, such as irrevocable water consumption, return water, transfer and others (see http://www.gostrf.com/normadata/1/4293791/4293791463.pdf). For Baikal, these components of the water balance are insignificant, therefore they are not included in the calculation. However, for the Sea of ​​Azov basin, these components are comparable to the inflow and therefore they must be taken into account. Adding any components of the water balance to equations (6) given in the article does not lead to a change in the optimization algorithm.

Point 3: Please check the attached file to see some other minor comments on the highlighted text.

Response 3: All comments and remarks except one were accepted and corrected. Answer to comment on line 494 "1.The level of the lake Baikal should be in the range [455.54, 457.5] m": “This requierement is canselled because 2 and 3”. Requirement 1 cannot be met, but 2 and 3 can . Requirements 1, 2 and 3 have different normative reliability (Figures 11, 12, 14 and others) and different priorities (for optimizing, these are penalty coefficients). Therefore, it was necessary to include all three criteria in optimization and in multi-criteria analysis. In addition, requirement 1 was adopted as a government decree.

Reviewer 2 Report

  1. The abstract should contain the summary of the analytical results, however, the author only talked about their method but haven't mentioned the final results and findings.
  2. Line 101, should be "are described in [1-3]"?
  3. Line 108, ","should be deleted?
  4. The author present a bunch of literatures and illustrated the short comes of these previous work. However, the advantage of this study has not been clearly described in the last part of introduction section (section 1). The author should stand out their improvement compared to other studies.
  5. In section 2, the detailed description of the datasets used in the study is missing, e.g., which institute provides the data, the spatiotemporal resolution of the dataset, etc.
  6. In the discussion part, could the authors compare their results with other similar studies? What are the new findings in this papers compared to others? What is the improvement of this study?

Author Response

Response to Reviewer 2 Comments

Point 1: The abstract should contain the summary of the analytical results, however, the author only talked about their method but haven't mentioned the final results and findings.

Response 1: The abstract supplemented by results and conclusions

Point 2: Line 101, should be "are described in [1-3]"?

Response 2: Line 101 (line 47 in the article) has been corrected.

Point 3: Line 108, ","should be deleted?

Response 3: Line 108 (line 54 in the article) has been corrected.

Point 4: The author present a bunch of literatures and illustrated the short comes of these previous work. However, the advantage of this study has not been clearly described in the last part of introduction section (section 1). The author should stand out their improvement compared to other studies.

Response 4: Section 1 "Introduction" is supplemented with a clear description of the benefits and results that distinguish the studies presented from those presented in the review.

Point 5: In section 2, the detailed description of the datasets used in the study is missing, e.g., which institute provides the data, the spatiotemporal resolution of the dataset, etc.

Response 5: Section "2. Materials and Methods" is supplemented with a detailed description and reference to the relevant sites of the used datasets location that are in the public domain. These are the observed historical data on the inflow to Lake Baikal, data on the modern requirements of water users, the main design parameters for waterworks. Also in the article the spatial-temporal resolution is given.

Point 6: In the discussion part, could the authors compare their results with other similar studies? What are the new findings in this papers compared to others? What is the improvement of this study?

Response 6: The authors compared their results with other similar studies (1. introduction). New conclusions were also formulated. To date, the authors have not found publications on the management of water resources of reservoirs in real time based on optimization in the form presented in the article (in our opinion, they do not exist at all). The presented algorithm, in the opinion of the authors, is the most promising, and of course it gives better results in comparison with the management according to the dispatch schedule. The authors placed the explanations in Section 5. Conclusions, and not in Section 4. Discussion, since, in our opinion, they fit better in Section 5.

Round 2

Reviewer 1 Report

Authors have addressed all my comments well therefore I suggest that the paper can be accepted.

Reviewer 2 Report

The authors addressed my comments, I recommend it to be published

This manuscript is a resubmission of an earlier submission. The following is a list of the peer review reports and author responses from that submission.

Round 1

Reviewer 1 Report

The introduction should include information about the problem, which should be investigated, background and state of art that explains the problem, as well as reasons for conducting the research.

The novelty of the paper should be explicitly highlighted. Deepen the discussion of results comparing with other case studies that address the same theme. Are there concrete steps that can be recommended?

What are the definitions for Reliability, Resilience, and Vulnerability?

Check the spelling in the whole manuscript, eg. reliabilty.

The manuscript should be carefully edited and checked.

Reviewer 2 Report

  1. The reviewed paper should be digested properly and summarized the key points related to the current study in a more precise way in the introduction.
  2. The flowcharts should be modified to convey a clearer picture of the study.
  3. As comments given in previous submission, the method, results , and discussion of the Bayesian method are not clear. What does the posterior density in Figure 5 mean? How to generate two 44-year forecast hydrological series after finding the change point? What’s the meaning of cut the observed inflow series in 1976? The generate inflows of 2020-2064 are not convincible, which make the reader difficult to understand the work of the authors. Also, the concept of the Schwab solar activity is even more confusing.
  4. How the compromise decisions are made and what’s the difference made by the proposed method are not clear in the manuscript.
  5. Why only finding the local optimum rather than global optimum?
  6. Moreover, it’s not clear why in tabular form of Figure 2 there are values like 7,7.2,…, 6 (and why tabular form is necessary in addition to graphic form) and why presenting time series for two time periods in the same figure is so important in this study (Figures 6-10).